# Building 1D lattice models with $G$-graded fusion category

**Shang-Qiang Ning[1,2], Bin-Bin Mao[2,3] and Chenjie Wang[2⋆]**

**1** Department of Physics, The Chinese University of Hong Kong,
Shatin, New Territories, Hong Kong, China
**2** Department of Physics and HKU-UCAS Joint Institute for Theoretical
and Computational Physics, The University of Hong Kong,
Pokfulam Road, Hong Kong, China
**3** School of Foundational Education,
University of Health and Rehabilitation Sciences,
Qingdao, China

⋆ cjwang@hku.hk

## Abstract

We construct a family of one-dimensional (1D) quantum lattice models based on $G$-graded unitary fusion category $\mathcal{C}_G$. This family realize an interpolation between the anyon-chain models and edge models of 2D symmetry-protected topological states, and can be thought of as edge models of 2D symmetry-enriched topological states. The models display a set of unconventional global symmetries that are characterized by the input category $\mathcal{C}_G$. While spontaneous symmetry breaking is also possible, our numerical evidence shows that the category symmetry constrains the models to the extent that the low-energy physics has a large likelihood to be gapless.

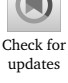

# 1 Introduction

It is hard to overstate the importance of symmetry in physics. Over the past decade, the role of symmetry has been extensively studied in topological states of matter, such as symmetry-protected topological (SPT) phases [1–3] and symmetry-enriched topological (SET) phases [4]. A lot of novel quantum states and phenomena are discovered by studying the interplay between symmetry and topology in quantum many-body systems.

The study of topological phases of matter in turn has advanced our understanding of symmetry. One of such advances is on 't Hooft anomaly of symmetry [5, 6]. 't Hooft anomaly is invariant under renormalization group flows, so it becomes a powerful tool to constrain the low-energy physics of a system. An anomalous system cannot admit a symmetric gapped non-degenerate ground state, but has to break symmetry spontaneously, or be gapless, or be topologically ordered (in two and higher dimensions) [7]. It is now understood that an anomalous system can be thought of as the boundary of an SPT bulk. In fact, for a given symmetry, 't Hooft anomalies are in one-to-one correspondence to SPT phases in one higher dimension. Because of the tremendous progress in the study of SPT phases in recent years, many new types of 't Hooft anomalies are discovered. One of the important instances is the famous Lieb-Shultz-Mattis theorem and its generalizations [8–10], which are actually consequences of 't Hooft anomalies involving lattice translation [11].

Recently people are interested in generalizing the concept of symmetry itself. Ordinary symmetries in quantum many-body systems are characterized by operators that act on the whole spatial manifold and form a group mathematically. One kind of generalized symmetries are $p$-form symmetries, which act on submanifolds of spatial co-dimension $p$ [12, 13]. For example, closed string operators associated with moving Abelian anyons in the 2D toric-code model are 1-form symmetries [14]. Another kind of generalized symmetries are non-invertible symmetries, whose corresponding operators form an algebra that does not admit a definition of inverse (i.e., beyond group). Non-invertible symmetries of a 1D system are naturally described by a fusion category [15–19]. In high dimensions, invertible and/or non-invertible symmetries of various co-dimensions collectively are characterized by higher fusion category, which itself is a subject still under development [20–26]. In this work, we will refer to all these generalized symmetries as *category symmetries*.

In fact, 't Hooft anomalies of ordinary finite symmetries can be well described within the language of category. For example, consider a 1D quantum system with a finite unitary symmetry group $G$. The 't Hooft anomalies are described by a 3-cocycle $v_3 : G \times G \times G \to U(1)$ [2]. The doublet $(G, v_3)$ forms a special fusion category, in which all simple objects are invertible. With this connection in mind, it is then not hard to understand that general category symmetries are also invariant under renormalization group flow and provide strong constraints on the low-energy physics of a theory [27]. Similar to conventional group-like symmetries, it is also possible to define anomaly-free and anomalous category symmetries [19, 27, 28]. In most of our discussions and statements, we implicitly assume that the category symmetries are anomalous, which are our main interests.

In this work, we pursue the idea of constraining low-energy physics with category symmetry in the particular context of building 1D quantum lattice models. A previous example of such lattice models is the Fibonacci anyon-chain model [29–31]. It describes a 1D array of interacting Fibonacci anyons, and has a generalized symmetry described by the Fibonacci fusion category. It turns out that the model is pinned at the tri-critical Ising conformal field theory (CFT) at low energy by the Fibonacci category symmetry. Classical counterparts of anyon-chain models are studied in Refs. [32, 33] and a recent on duality of category symmetry and extension to module category is given in Ref. [34] using the framework of the tensor-network states. Another family of such 1D lattice models are the effective edge theory of 2D SPT lattice models, e.g., those in Refs. [35–38]. These models respect a non-onsite symmetry group $G$ with a nontrivial 3-cocycle $v_3$, or equivalently, a category symmetry $\mathcal{C} = (G, v_3)$. It is found that the low-energy physics of these models in a very large parameter space are gapless CFTs (spontaneous symmetry breaking is another possibility).

We construct a family of 1D quantum lattice models based on a general $G$-graded unitary fusion category (UFC) $\mathcal{C}_G$. A fusion category equipped with a $G$-grading structure has a decomposition $\mathcal{C}_G = \bigoplus_{\mathbf{g} \in G} \mathcal{C}_{\mathbf{g}}$, with $G$ being a finite group (see Sec. 2.1). In our model, $\mathcal{C}_G$ serves both as the input data and as the characterization of symmetries. We start by building a 1D lattice Hilbert space out of $\mathcal{C}_G$, which in general does not have a tensor-product structure. The language of fusion category allows us to naturally associate every object in $\mathcal{C}_G$ with an operator, which we will use as symmetry operator. Then, we design a minimal Hamiltonian that commutes with these symmetry operators. It turns out that our model unifies the anyon chain model [29] and edge model of 2D bosonic SPTs [37]. When $G$ is trivial, it reduces to the anyon chains; when $\mathcal{C}_0$ is trivial ("0" denotes the identity of $G$), i.e., $\mathcal{C}_G = (G, v_3)$, it reduces to the SPT edge model (our model is slightly more general than Ref. [37] by having more parameters). Therefore, our model provides an interpolation between the anyon-chain model and the SPT edge model. For general $\mathcal{C}_G$, we find that our model can be thought of as a boundary theory of 2D SET models (under an appropriate boundary condition) [39, 40].

We have numerically studied the low-energy physics of a few examples of our model. As mentioned above, we are mainly interested in anomalous category symmetries. A sufficient condition for a category to be anomalous is that it contains objects with non-integer quantum dimensions [27], and most of our examples satisfy this condition. In the example of $\mathcal{C}_G = (\mathbb{Z}_2, v_3)$ with $v_3$ being the nontrivial 3-cocycle, the phase diagram shows an *extended* quantum critical region in the parameter space which are characterized by Luttinger liquids (Fig. 4). When $\mathcal{C}_G$ is the Ising fusion category (Sec. 3.3), we find that the low-energy physics is characterized by the critical Ising CFT at certain choices of parameters (this example is identical to that in Ref. [38]). For the $\mathbb{Z}_3$ Tambara-Yamagami category (Sec. 3.4), we find the low-energy physics is described by the critical 3-state Potts CFT. While more numerical effort is needed for investigating the whole phase diagram of the latter examples, our current results have already demonstrated that anomalous category symmetry $\mathcal{C}_G$ constrains the model to the extent that the low-energy physics has a large likelihood to sit at quantum criticality. We note

that Ref. [41] has constructed a class of exactly solvable models with fusion category symmetry. The fusion category symmetry of these models is non-anomalous, so the models admit a symmetric gapped non-degenerate ground state.

The rest of the paper is outlined as follows. In Sec. 2, we build up the model. After introducing some basic knowledge of $G$-graded UFC in Sec. 2.1, we construct the Hilbert space in Sec. 2.2, write out explicit expressions of the symmetry operators in Sec. 2.3, and construct a minimal symmetric Hamiltonian in Sec. 2.4. We then present a few examples of our model in Sec. 3, including the two limiting cases ($G$ being trivial and $\mathcal{C}_0$ being trivial), Ising fusion category, Tambara-Yamagami category, etc. We also present some numerical results in Sec. 3.5. We discuss the issue of the gauge choice of $F$ symbol and its consequence to the model in Sec. 4.1, and the relation of our model to the boundary of SET models in Sec. 4.2. In Sec. 5, we make a summary and discuss a few future directions. Appendices include some technical details.

## 2 Model

In this section, we describe the model. We begin with some basics of $G$-graded unitary fusion category, which describes the input data of the model. The Hilbert space is constructed out of fusion spaces of a $G$-graded UFC, which, in general, does not admit a tensor product structure. Then, we write down a series of generalized symmetries and construct a general minimal Hamiltonian that respects these symmetries. The generalized symmetries are characterized by the input category $\mathcal{C}_G$ too.

### 2.1 Basics of $G$-graded fusion category

The input data of our model is a $G$-graded unitary fusion category $\mathcal{C}_G$ [42, 43], where $G$ is a finite group. A category $\mathcal{C}_G$ contains a finite list of simple objects,[1] denoted as $a$, $b$, $c$, etc. Composite objects are written as a formal sum of simple objects $\sum_a n_a a$, with $n_a$ a non-negative integer. Simple objects follow a set of fusion rules $a \times b = \sum_c N_c^{ab} c$, where the integer $N_c^{ab} \geq 0$ is called fusion multiplicity. In general, fusion rules are not commutative, i.e. $a \times b \neq b \times a$. There exists a special object $\mathbb{1}$, called the identity or vacuum, satisfying $\mathbb{1} \times a = a \times \mathbb{1} = a$ for any $a$. Every simple object comes with a quantum dimension $d_a$, which satisfies $d_a d_b = \sum_c N_{ab}^c d_c$. $D = \sqrt{\sum_a d_a^2}$ is called the total quantum dimension. Every fusion channel $c$ in $a \times b$ with $N_{ab}^c \neq 0$ is associated with a vector space $\mathbb{V}_c^{ab}$ of dimension $N_{ab}^c$, called the fusion space. The basis state $|ab; c, \mu\rangle \in \mathbb{V}_c^{ab}$ can be graphically represented as

$$|ab; c, \mu\rangle = \vcenter{\hbox{\includegraphics{eq1}}} . \tag{1}$$

An important quantity of $\mathcal{C}_G$ is the $F$ symbol, which is an isomorphism $F_d^{abc} : \bigoplus_e \mathbb{V}_e^{ab} \otimes \mathbb{V}_d^{ec} \to \bigoplus_f \mathbb{V}_d^{af} \otimes \mathbb{V}_f^{bc}$. With the basis vectors, it is given by

$$\vcenter{\hbox{\includegraphics{eq2a}}} = \sum_{f\alpha\beta} \left(F_d^{abc}\right)_{e\mu\nu}^{f\alpha\beta} \vcenter{\hbox{\includegraphics{eq2b}}} . \tag{2}$$

---

[1] If a fusion category is braided, simple objects correspond to anyons in two-dimensional topological order. In our model, a braiding structure in $\mathcal{C}_G$ is not required.

(a)

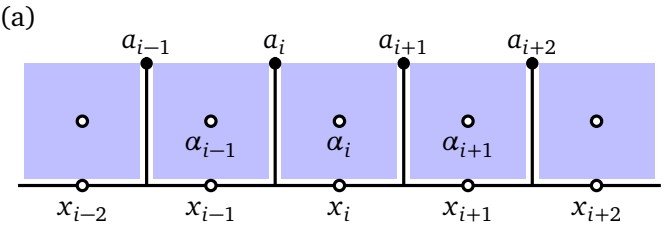

(b)

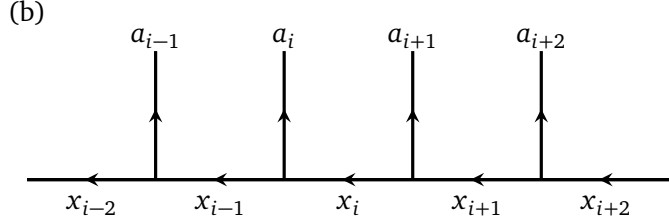

Figure 1: (a) Lattice of our 1D model. Blue regions are viewed as domains, and lines are viewed as domain walls. Each unit cell contains two dynamical variables $\alpha_i$ and $x_i$ (empty circle), and a slaved variable $a_i$ (black dot). The "domain" variable $\alpha_i$ is an element of a finite group $G$. The empty region below the horizontal line is viewed as the domain associated with the identity of $G$. A given configuration $\{\alpha_i\}$ fixes the "domain wall" variables $\{a_i\}$. Each $a_i$ is a pre-selected object in $\mathcal{C}_{\alpha_{i-1}^{-1}\alpha_i} \subset \mathcal{C}_G$, where $\mathcal{C}_G$ is a $G$-graded fusion category. The second dynamical variable $x_i \in \mathcal{C}_{\alpha_i}$ is a fusion channel of $x_{i-1} \times a_i$. Every valid configuration $\{\alpha_i, x_i\}$ gives a quantum state $|\{\alpha_i, x_i\}\rangle$, which all together form a basis of the lattice model. (b) The domain wall lines form a fusion tree of the objects $\{a_i\}$.

Since we can perform basis transforms in $\mathbb{V}_c^{bc}$, the $F$ symbols depend choices of basis. In addition, they also satisfy consistency conditions, known as the pentagon equations [42]. Throughout the paper, we assume that $\mathcal{C}_G$ is multiplicity-free, i.e., $N_{ab}^c = 0$ or $1$, for simplicity. Accordingly, the index $\mu$ in (1) is not needed.

The above properties are true for any unitary fusion category. The $G$-grading structure means that $\mathcal{C}_G$ has the following decomposition

$$\mathcal{C}_G = \bigoplus_{\mathbf{g}\in G} \mathcal{C}_{\mathbf{g}}, \tag{3}$$

with $\mathbb{1} \in \mathcal{C}_0$.[2] If $a \in \mathcal{C}_{\mathbf{g}}$, we will often denote it as $a_{\mathbf{g}}$. The grading structure is respected by fusion, $a_{\mathbf{g}} \times b_{\mathbf{h}} = \sum_{c_{\mathbf{gh}}} N_{ab}^c c_{\mathbf{gh}}$. Given a set of $F$ symbols $F^{a_{\mathbf{g}} b_{\mathbf{h}} c_{\mathbf{k}}}$, we can modify it to obtain a new $G$-graded fusion category $\tilde{\mathcal{C}}_G$ as follows

$$\tilde{F}^{a_{\mathbf{g}} b_{\mathbf{h}} c_{\mathbf{k}}} = F^{a_{\mathbf{g}} b_{\mathbf{h}} c_{\mathbf{k}}} \nu_3(\mathbf{g}, \mathbf{h}, \mathbf{k}), \tag{4}$$

where $\nu_3(\mathbf{g}, \mathbf{h}, \mathbf{k})$ is a 3-cocycle of $G$. If we define $D_{\mathbf{g}} = \sqrt{\sum_{a\in\mathcal{C}_{\mathbf{g}}} d_a^2}$, then $D_{\mathbf{g}} = D_0$ for all $\mathbf{g}$. Then, $D = D_0\sqrt{|G|}$.

Such $G$-graded fusion categories naturally appear in the study of SET phases. For more details of unitary fusion categories, readers may consult Ref. [16, 42, 43]. For our purpose of constructing models, we will need the set of simple objects $\{a\}$, fusion rules described by $\{N_{ab}^c\}$, explicit expressions of $F$ symbols, and the $G$-grading structure.

---

[2]We use either 0 or 1 to denote the identity of $G$ depending on the context.

## 2.2 Hilbert space

The Hilbert space $\mathcal{H}$ of our model is defined on a 1D lattice of length $L$, shown in Fig. 1. It has the following structure

$$\mathcal{H} = \bigoplus_{\{\alpha_i\}} \mathcal{H}^{\text{fusion}}_{\{\alpha_i\}}, \tag{5}$$

where $\alpha_i \in G$ is a "domain" variable in the $i$th unit cell, and $\mathcal{H}^{\text{fusion}}_{\{\alpha_i\}}$ is the fusion space of objects $\{a_i\}$ with $a_i \in \mathcal{C}_G$. The set $\{a_i\}$ is determined by the domain configuration $\{\alpha_i\}$ as follows: each $\{\alpha_i\}$ defines a series of "domain walls" labeled by $\mathbf{g}_i = \alpha_{i-1}^{-1} \alpha_i$ (vertical lines in Fig. 1a), and an object $a_i \in \mathcal{C}_{\mathbf{g}_i}$ is then picked out and put on the $i$th domain wall. We pre-select a particular object $a_{\mathbf{g}} \in \mathcal{C}_{\mathbf{g}}$ for every $\mathbf{g}$, such that $a_i$ is determined by $\mathbf{g}_i$ via $a_i = a_{\mathbf{g}_i}$. Let $\mathcal{A} = \{a_{\mathbf{g}} | \forall \mathbf{g} \in G\}$ be the collection of selected objects. Then, the triplet $(G, \mathcal{C}_G, \mathcal{A})$ defines the Hilbert space $\mathcal{H}$.

Let $\{x_i\}$ be the possible fusion channels of $\{a_i\}$. The space $\mathcal{H}^{\text{fusion}}_{\{\alpha_i\}}$ is spanned by fusion states of $\{a_i\}$, pictorially described by Fig. 1b. To avoid ambiguity, we take $x_i \in \mathcal{C}_{\alpha_i}$. This corresponds to the choice that the empty region below the horizontal line in Fig. 1a is viewed as the identity domain, i.e., $\alpha_{\text{empty}} = 1$. Accordingly, $x_i$ is the domain wall between $\alpha_{\text{empty}}$ and $\alpha_i$. Combining domain variables $\{\alpha_i\}$ and fusion channels $\{x_i\}$, we denote the basis vectors of $\mathcal{H}$ as $|\{\alpha_i, x_i\}\rangle$. In most part of the paper, we assume periodic boundary conditions.

A few remarks are in order. First, in general, $\mathcal{H}$ does not have a tensor-product structure. In the special case that $\mathcal{C}_0 = \{\mathbb{1}\}$, $\mathcal{H}^{\text{fusion}}_{\{\alpha_i\}}$ is one-dimensional. This makes $\mathcal{H}$ a tensor-product vector space, $\mathcal{H} = \bigotimes_i \mathbb{V}_i^G$, where $\mathbb{V}_i^G = \text{span}\{|\alpha_i\rangle | \alpha_i \in G\}$. Second, we have selected a subset $\mathcal{A} \subset \mathcal{C}_G$ when building up the Hilbert space. Physically, we view objects in $\mathcal{C}_{\mathbf{g}}$ as different topological defects that can live on a $\mathbf{g}$ domain wall. Those defects in $\mathcal{A}$ are selected *by hand* in the current construction. Alternatively, one may allow $a_i$ to vary in $\mathcal{C}_{\mathbf{g}_i}$ and add a term in the Hamiltonian to select the particular defect $a_{\mathbf{g}} \in \mathcal{A}$ energetically (see a discussion around Eq. (71) in Sec. 4.2). However, this will make the Hilbert space larger and less friendly for numerical calculations. Third, if $\mathcal{C}_G$ has nontrivial fusion multiplicities, one needs to include another variable $\mu_i = 1, \ldots, N^{x_i}_{x_{i-1} a_i}$ at the vertex associated with fusing $x_{i-1}$ and $a_i$ into $x_i$. It is neglected in our construction as we always assume that $\mathcal{C}_G$ is multiplicity-free.

## 2.3 Category symmetry

An advantage of using the fusion category language to build up the Hilbert space is that it helps to naturally define a set of operators which will serve as symmetry operators in our model. An interesting feature is that these operators follow the fusion algebra of $\mathcal{C}_G$ [Eq. (7)], which in general is not group-multiplication-like. Such kind of symmetries are called different names in the literature, e.g., algebraic symmetry, categorical symmetry or non-invertible symmetry. We will simply call them *category symmetry*, as opposed to the usual group symmetry. Even if in the special case that $\mathcal{C}_0 = \{\mathbb{1}\}$ and the fusion algebra associated with $\mathcal{C}_G$ reduces to group multiplication of $G$, we will see that the symmetry group $G$ carries a 't Hooft anomaly in general due to nontrivial $F$ symbols. It implies that our model is not featureless in general, but has to either break symmetries or be gapless.

For each simple object $y_{\mathbf{h}} \in \mathcal{C}_G$, we can write down a symmetry operator $U(y_{\mathbf{h}})$. Under the action of $U(y_{\mathbf{h}})$, the domain variable $\alpha_i$ is mapped $\mathbf{h}\alpha_i$, simultaneously for every $i$. This leaves the domain wall $\mathbf{g}_i = \alpha_{i-1}^{-1} \alpha_i$ unchanged, so does the defect $a_i$ on it. The action on the fusion channels is associated with the matrix element

$$\langle \{\mathbf{h}\alpha_i, x_i'\} | U(y_{\mathbf{h}}) | \{\alpha_i, x_i\}\rangle = \prod_{i=1}^{L} \left[ (F^{y_{\mathbf{h}}, x_i, a_{i+1}}_{x_{i+1}'})^\dagger \right]^{x_i'}_{x_{i+1}}, \tag{6}$$

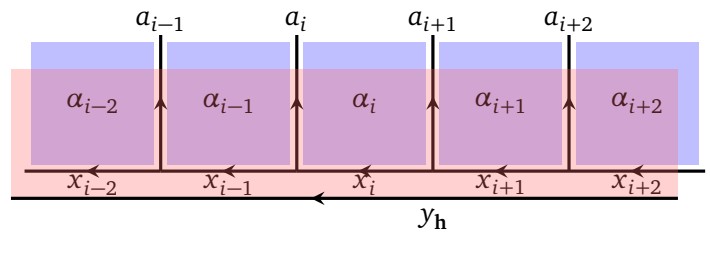

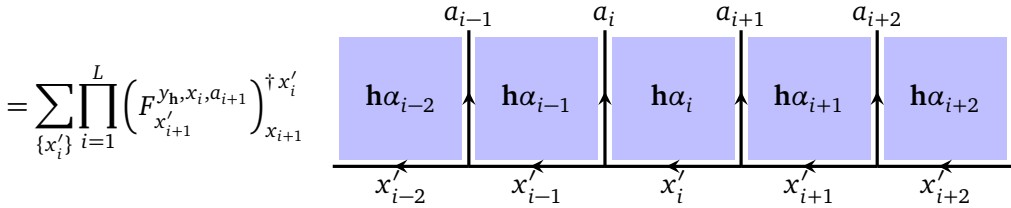

Figure 2: Graphical representation of $U(y_\mathbf{h})$. The equation is obtained by fusing a uniform $\mathbf{h}$ domain onto $\{\alpha_i\}$, and a $y_\mathbf{h}$ line into $\{x_i\}$.

where $x_i \in \mathcal{C}_{\alpha_i}$ and $x_i' \in \mathcal{C}_{\mathbf{h}\alpha_i}$. The matrix element $\langle\{\alpha_i', x_i'\}|U(y_\mathbf{h})|\{\alpha_i, x_i\}\rangle = 0$, if $\alpha_i' \neq \mathbf{h}\alpha_i$. The operator $U(y_\mathbf{h})$ has a graphical representation, shown in Fig. 2: it is represented by fusing a uniform $\mathbf{h}$ domain and its $y_\mathbf{h}$ domain wall with respect to the vacuum into the state $|\{\alpha_i, x_i\}\rangle$. We show in Appendix A that the fusion process indeed gives Eq. (6).

The symmetry operators satisfy the algebraic relation

$$U(x_\mathbf{g})U(y_\mathbf{h}) = \sum_{z_\mathbf{k}} N^{z_\mathbf{k}}_{x_\mathbf{g}y_\mathbf{h}} U(z_\mathbf{k}). \tag{7}$$

This relation follows directly from that fusion processes are associative and the fusion rule is given by $x_\mathbf{g} \times y_\mathbf{h} = \sum_{z_\mathbf{k}} N^{z_\mathbf{k}}_{x_\mathbf{g}y_\mathbf{h}} z_\mathbf{k}$. One can also use Eq. (6) to explicitly check this algebra. As studied in many previous works, this kind of algebraic symmetries can help (although not guarantee) a lattice model to sit at quantum criticality. We will demonstrate this when we discuss examples in Sec. 3.

## 2.4 Hamiltonian

With the set of symmetries $U(y_\mathbf{h})$ in hand, we would like to write down a "minimal" Hamiltonian that respects these symmetries. We will consider a Hamiltonian of the form

$$H = -\sum_i H_i, \tag{8}$$

and require $H_i$ to be an operator that acts only on the $(i-1)$th, $i$th and $(i+1)$th unit cells. To define $H_i$, it is convenient to work in an alternative basis. The alternative basis is related to the original basis through an $F$ move as follows:

$$\left| \begin{matrix} & a_i & a_{i+1} \\ \alpha_{i-1} & \alpha_i & \alpha_{i+1} \\ & & \\ x_{i-1} & x_i & x_{i+1} \end{matrix} \right\rangle = \sum_{z_i} \left[ F^{x_{i-1}a_ia_{i+1}}_{x_{i+1}} \right]^{z_i}_{x_i} \left| \begin{matrix} a_i & & a_{i+1} \\ & \alpha_i & \\ \alpha_{i-1} & & \alpha_{i+1} \\ & z_i & \\ x_{i-1} & & x_{i+1} \end{matrix} \right\rangle , \tag{9}$$

where $z_i$ runs over all outcomes in the fusion product $a_i \times a_{i+1}$. The term $H_i$ in the new basis is given by

$$\left\langle \begin{array}{c} a'_i \quad a'_{i+1} \\ \alpha'_{i-1} \quad \alpha'_i \quad z'_i \quad \alpha'_{i+1} \\ x'_{i-1} \quad x'_{i+1} \end{array} \middle| H_i \middle| \begin{array}{c} a_i \quad a_{i+1} \\ \alpha_{i-1} \quad \alpha_i \quad z_i \quad \alpha_{i+1} \\ x_{i-1} \quad x_{i+1} \end{array} \right\rangle = w^{z_i}_{\alpha_i^{-1}\alpha'_i} \delta^{\alpha'_{i-1}}_{\alpha_{i-1}} \delta^{\alpha'_{i+1}}_{\alpha_{i+1}} \delta^{x'_{i-1}}_{x_{i-1}} \delta^{x'_{i+1}}_{x_{i+1}} \delta^{z'_i}_{z_i}, \qquad (10)$$

where $\delta^{a'}_a = 1$ if $a = a'$, and $\delta^{a'}_a = 0$ otherwise. That is, $H_i$ only flips the domain variable $\alpha_i$ to $\alpha'_i$, with the transition amplitude denoted as $w^{z_i}_{\alpha_i^{-1}\alpha'_i}$. We assume the transition amplitude only depends on the domain shift $\mathbf{h}_i = \alpha_i^{-1}\alpha'_i$ and the fusion channel $z_i$. One may consider a more complicated transition amplitude. However, we find that the current choice is already enough to produce interesting results. Hermiticity requires that $w^z_{\mathbf{h}^{-1}} = (w^z_{\mathbf{h}})^*$.

Using the transformation (9), the nonzero matrix elements of $H_i$ in the original basis are given by

$$\left\langle \begin{array}{c} a'_i \quad a'_{i+1} \\ \alpha_{i-1} \quad \alpha'_i \quad \alpha_{i+1} \\ x_{i-1} \quad x'_i \quad x_{i+1} \end{array} \middle| H_i \middle| \begin{array}{c} a_i \quad a_{i+1} \\ \alpha_{i-1} \quad \alpha_i \quad \alpha_{i+1} \\ x_{i-1} \quad x_i \quad x_{i+1} \end{array} \right\rangle = \sum_{z_i} w^{z_i}_{\alpha_i^{-1}\alpha'_i} \left[ \left( F^{x_{i-1}a'_i a'_{i+1}}_{x_{i+1}} \right)^\dagger \right]^{x'_i}_{z_i} \left( F^{x_{i-1}a_i a_{i+1}}_{x_{i+1}} \right)^{z_i}_{x_i},$$

$$(11)$$

where the sum runs over those $z_i$'s that are simultaneously in $a_i \times a_{i+1}$ and $a'_i \times a'_{i+1}$. Note that $F$ symbols can be zero for certain choices of $z_i$ due to incompatible fusion. Our model is a natural generalization of the anyon fusion chain model first proposed in Ref. [29].

The Hamiltonian $H$ is symmetric under the category symmetry $U(y_{\mathbf{h}})$ in (6). An easy way to see this is through Eq. (10). In that expression, $H_i$ is independent of $x_{i-1}$ and $x_i$ and diagonal in the variable $z_i$. Meanwhile, $U(y_{\mathbf{h}})$ corresponds to flipping all $\alpha_i$ and fusing a $y_{\mathbf{g}}$ string, which does not change $z_i$ and only flips $x_{i-1}$ and $x_{i+1}$. It is clear that the action of $H_i$ and $U(y_{\mathbf{h}})$ commute. For a more explicit derivation, readers are referred to Appendix A.

We remark that $F$ symbols depend on gauge choices. Since Eq. (11) explicitly depends on $F$, our model has an explicit dependence on the gauge choice. Below we mainly focus on examples with gauge-inequivalent $F$ symbols. We discuss some implications of gauge choices of $F$ in Sec. 4.1.

## 3 Examples

The model defined in Eqs. (8) and (11) provides an "interpolation" of the anyon-chain model [29] and the SPT edge model [37]. The latter two are special cases of our model. More generally, our model can be thought of as an edge model of 2D SET phases (see Sec. 4.2). Below we discuss how it is related to anyon chains and SPT edge models, and explore a few interesting examples with numerical calculations.

### 3.1 Anyon chain

When $G$ is trivial, $\mathcal{C}_G = \mathcal{C}_0$. Then, our model reduces to the well-known anyon chain model. In this case, there is no domain variable, i.e., $\alpha_i = 0$. On domain walls, every $a_i$ is set to be a simple object $a \in \mathcal{C}_0$. The Hamiltonian (11) then reads

$$\langle x_{i-1} x'_i x_i | H_i | x_{i-1} x_i x_{i+1} \rangle = \sum_z w^z \left[ \left( F^{x_{i-1}aa}_{x_{i+1}} \right)^\dagger \right]^{x'_i}_z \left( F^{x_{i-1}aa}_{x_{i+1}} \right)^z_{x_i}, \qquad (12)$$

where the site dependence of $z_i$ is dropped since all $a_i$'s are identical. The coefficient $w^z \equiv w_0^z$ is the energy of the fusion channel $z$ of two neighboring $a$'s. This is exactly the anyon-chain Hamiltonian that has been widely studied, e.g., in Refs. [29–31]. The simplest example is the golden chain model, with $\mathcal{C}_0$ being the fusion category of Fibonacci anyons. It was found the the category symmetries $\{U(y)\}$ enforce the anyon-chain model to sit at quantum criticality [29, 44] or to break symmetry spontaneously.

## 3.2 Edge model of bosonic SPTs

Another limit of our model is $\mathcal{C}_0 = \{\mathbb{1}\}$. In this case, $\mathcal{C}_G$ is equivalent to the doublet $(G, v_3)$, where $v_3 = v_3(\mathbf{g}, \mathbf{h}, \mathbf{k}) \in \mathcal{H}^3(G, U(1))$ is a 3-cocycle. There is only one simple object in each $\mathcal{C}_{\mathbf{g}}$, and the fusion algebra of $\mathcal{C}_G$ reduces to group multiplication of $G$. We use the group element $\mathbf{g}$ to denote the simple object in $\mathcal{C}_{\mathbf{g}}$. It has $d_{\mathbf{g}} = 1$. The $F$ symbol is determined by $v_3$, $(F_{\mathbf{ghk}}^{\mathbf{g,h,k}})_{\mathbf{gh}}^{\mathbf{hk}} = v_3(\mathbf{g}, \mathbf{h}, \mathbf{k})$. This kind of $G$-graded fusion category appears in the study of symmetry defects in 2D bosonic SPT phases with symmetry group $G$ [2]. Below we will see that our model can be viewed as an effective edge model for 2D bosonic SPT phases.

Since all simple objects in $\mathcal{C}_G$ have quantum dimension 1, the Hilbert space has a tensor-product structure, $\mathcal{H} = \bigotimes_i \mathbb{V}_i^G$, where $\mathbb{V}_i^G = \mathrm{span}\{|\alpha_i\rangle | \alpha_i \in G\}$. Given a domain configuration, $\{a_i\}$ and $\{x_i\}$ are uniquely determined, with $a_i = \alpha_{i-1}^{-1}\alpha_i$ and $x_i = \alpha_i$. Then, our model reduces to

$$\langle \alpha_{i-1}\alpha_i'\alpha_{i+1}|H_i|\alpha_{i-1}\alpha_i\alpha_{i+1}\rangle = w_{\mathbf{h}_i}^{z_i} \frac{v_3(\alpha_{i-1}, \alpha_{i-1}^{-1}\alpha_i, \alpha_i^{-1}\alpha_{i+1})}{v_3(\alpha_{i-1}, \alpha_{i-1}^{-1}\alpha_i', (\alpha_i')^{-1}\alpha_{i+1})}, \tag{13}$$

where $z_i = \alpha_{i-1}^{-1}\alpha_{i+1}$ and $\mathbf{h}_i = \alpha_i^{-1}\alpha_i'$. If we take $w_{\mathbf{h}}^z = 1$ for every $\mathbf{h}$ and $z$, the model reduces to the SPT domain-wall model of Ref. [37], which was derived by considering a domain wall of two 2D SPT models and projecting out the bulk degrees of freedom.[3] It was shown numerically there that for various choices of $G$ and $v_3$, the low-energy spectrum is gapless and described by a conformal field theory with an integer central charge (i.e., a Luttinger liquid). For more general $w_{\mathbf{h}}^z$, we will also give numerical evidence in Sec. 3.5 that the model is gapless and quantum critical in an extended region of the parameter space, by considering the example $G = \mathbb{Z}_2$ (see Fig. 4).

The model carries a 't Hooft anomaly of the symmetry group $G$. For $\mathcal{C}_G = (G, v_3)$, the symmetry operator $U(y_{\mathbf{g}}) \equiv U(\mathbf{g})$ in (6) becomes

$$\langle \mathbf{g}\alpha_1, ..., \mathbf{g}\alpha_L|U(\mathbf{g})|\alpha_1, ..., \alpha_L\rangle = \prod_{i=1}^{L} v_3^*(\mathbf{g}, \alpha_i, \alpha_i^{-1}\alpha_{i+1}). \tag{14}$$

The symmetry algebra (7) reduces to the multiplication of group elements in $G$. While the Hilbert space has a tensor-product structure, this particular realization $\{U(\mathbf{g})\}$ of symmetry group $G$ is not *onsite*, making it to carry a 't Hooft anomaly. The anomaly can be extracted through the procedure proposed in Ref. [45], which we find is precisely the 3-cocycle $v_3$. According to bulk-boundary correspondence, this model cannot be realized on a 1D lattice if we insist $G$ to be realized in an onsite way (when $v_3$ is a nontrivial cocycle). Onsite realizations can only be achieved at the edge of a 2D SPT bulk characterized by the 3-cocycle $v_3 \in \mathcal{H}^3(G, U(1))$ (Sec. 4.2 gives an explicit discussion of the edge viewpoint). Therefore, our 1D model mimics the edge of a 2D bosonic SPT bulk by sacrificing the onsiteness of the symmetry operators.

---

[3]To make an exact match, the 3-cocycle $v_{ab}$ in Eq. (40) of Ref. [37] is related to our 3-cocycle by $v_{ab}(\alpha_1, \alpha_2, \alpha_3) = v_3^*(\alpha_3^{-1}, \alpha_2^{-1}, \alpha_1^{-1})$. One also needs to convert the homogeneous cocycle in Ref. [37] to inhomogeneous cocycle and set the parameter $g^* = 1$ there.

As is known, such a 1D system cannot be featureless (i.e., gapped and symmetric with a non-degenerate ground state). While we cannot rule out spontaneous symmetry breaking, the 't Hooft anomaly does increase the likelihood of being gapless.

### 3.2.1 $G = \mathbb{Z}_2$

After the above general remarks, we now take a close look at the $\mathbb{Z}_2$ case. Taking $\mathbb{Z}_2 = \{0, 1\}$ with an additive group multiplication, we have four real parameters in the Hamiltonian (13): $w_0^0$, $w_0^1$, $w_1^0$ and $w_1^1$. The cohomology group $H^3(\mathbb{Z}_2, U(1)) = \mathbb{Z}_2$, so there are two inequivalent classes of $\nu_3$. An explicit expression of $\nu_3$ is given by

$$\nu_3(a, b, c) = (-1)^{kabc}, \tag{15}$$

where $a, b, c = 0, 1$ are group elements of $\mathbb{Z}_2$. When $k = 0$, $\nu_3$ is trivial. When $k = 1$, $\nu_3$ is nontrivial.

Let us take $\alpha_i = \pm 1$ to represent $\mathbb{Z}_2$ and rewrite the Hamiltonian (13) with Pauli matrices. Let $s_i^x$, $s_i^y$ and $s_i^z$ be the Pauli matrices. It is straightforward to show that, for the trivial $\nu_3$,

$$H_i^0 = \frac{w_0^0 - w_0^1}{2} s_{i-1}^z s_{i+1}^z + \frac{1}{2}\left[w_1^0(1 + s_{i-1}^z s_{i+1}^z) + w_1^1(1 - s_{i-1}^z s_{i+1}^z)\right]s_i^x, \tag{16}$$

and for the nontrivial $\nu_3$,

$$H_i^1 = \frac{w_0^0 - w_0^1}{2} s_{i-1}^z s_{i+1}^z + \frac{1}{2}\left[w_1^0(s_{i-1}^z + s_{i+1}^z) + w_1^1(1 - s_{i-1}^z s_{i+1}^z)\right]s_i^x, \tag{17}$$

where a constant term $(w_0^0 + w_0^1)/2$ has been omitted in both $H_i^0$ and $H_i^1$. The symmetry operator for the nontrivial $\mathbb{Z}_2$ group element can be written as

$$U^0 = \prod_i s_i^x, \qquad U^1 = e^{i\pi \sum_i (1 - s_i^z s_{i+1}^z)/4} \prod_i s_i^x, \tag{18}$$

for the two models, respectively. Note that the term $\sum_i (1 - s_i^z s_{i+1}^z)$ is always a multiple of 4 under periodic boundary conditions. Also note that the two models are identical when $w_1^0 = 0$. When $w_0^0 = w_0^1$, the model $H^1 = -\sum_i H_i^1$ is exactly the Ising domain wall model in Ref. [37] derived from the interface between 2D SPT bulks.

Let us introduce the following re-parametrization,

$$J = w_0^1 - w_0^0, \qquad \Delta = \sqrt{(w_1^0)^2 + (w_1^1)^2}, \qquad w_1^0 = \Delta\cos\theta, \qquad w_1^1 = \Delta\sin\theta. \tag{19}$$

Then, the two Hamiltonians $H^a = -\sum_i H_i^a$ ($a = 0, 1$) can be written as

$$H^0 = \frac{J}{2}\sum_i s_{i-1}^z s_{i+1}^z - \frac{\Delta}{2}\sum_i \left[\cos\theta(1 + s_{i-1}^z s_{i+1}^z) + \sin\theta(1 - s_{i-1}^z s_{i+1}^z)\right]s_i^x, \tag{20}$$

and

$$H^1 = \frac{J}{2}\sum_i s_{i-1}^z s_{i+1}^z - \frac{\Delta}{2}\sum_i \left[\cos\theta(s_{i-1}^z + s_{i+1}^z) + \sin\theta(1 - s_{i-1}^z s_{i+1}^z)\right]s_i^x. \tag{21}$$

We define the dimensionless parameter $r = J/\Delta$. The phase diagrams of $H^0$ and $H^1$ will be plotted in the $(r, \theta)$ plane.

The model $H^0$ can be mapped to the usual XYZ model by the Kramers-Wannier duality: $s_{i-1}^z s_i^z = \mu_i^x$ and $s_i^x = \mu_i^z \mu_{i+1}^z$. With the mapping, we have

$$H^0 = -\sum_i (J_x \mu_i^x \mu_{i+1}^x + J_y \mu_i^y \mu_{i+1}^y + J_z \mu_i^z \mu_{i+1}^z), \tag{22}$$

where

$$J_x = -\frac{J}{2}, \quad J_y = \frac{\Delta(\sin\theta - \cos\theta)}{2}, \quad J_z = \frac{\Delta(\sin\theta + \cos\theta)}{2}. \tag{23}$$

The phase diagram of XYZ model is known [46]. It is gapless if and only if the condition $|J_x| = |J_y| \geq |J_z|$ or its cyclic permutation is satisfied; otherwise, it is gapped with magnetic ordering. The gapless condition leads to the phase diagram in Fig. 3. Let us take the $\theta = 0$ critical line for example. It reduces to the XXZ model. When $|r| > 1$, the model is in a magnetically ordered phase, i.e, a spontaneous symmetry breaking phase. When $|r| < 1$, the model is a Luttinger liquid, with the Luttinger liquid parameter $K \in (\frac{1}{2}, \infty)$. At the transition point $r = -1$, the model is equivalent to the $SO(3)$-symmetric anti-ferromagnetic Heisenberg chain, whose low-energy physics is a Luttinger liquid with $K = 1/2$, or equivalently, the $SU(2)_1$ conformal field theory. At the transition point $r = 1$, the Luttinger liquid parameter $K \to \infty$ and the low-energy spectrum has a quadratic dispersion in momentum. Other critical lines in the phase diagram are similar.

To study the phase diagram of $H^1$, we first perform a unitary transformation $H^1 \to SH^1S^\dagger$ with $S = \prod_j e^{i\pi s_j^z s_{j+1}^z/8 + i(\pi - 2\theta)s_j^z/4}$. After the transformation, the new Hamiltonian reads

$$H^1 = \frac{J}{2}\sum_i s_{i-1}^z s_{i+1}^z + \frac{\Delta}{2}\sum_i (\cos 2\theta s_i^x + \sin 2\theta s_i^y + s_{i-1}^z s_i^x s_{i+1}^z). \tag{24}$$

Accordingly, the phase diagram is symmetric under the shifting $\theta \to \theta + \pi$. In addition, under the transformation $S' = \prod_i s_i^x$, the Hamiltonian $H^1(\theta) \to H^1(-\theta)$. Therefore, it is enough to study the phase diagram for $\theta \in [0, \pi/2]$.

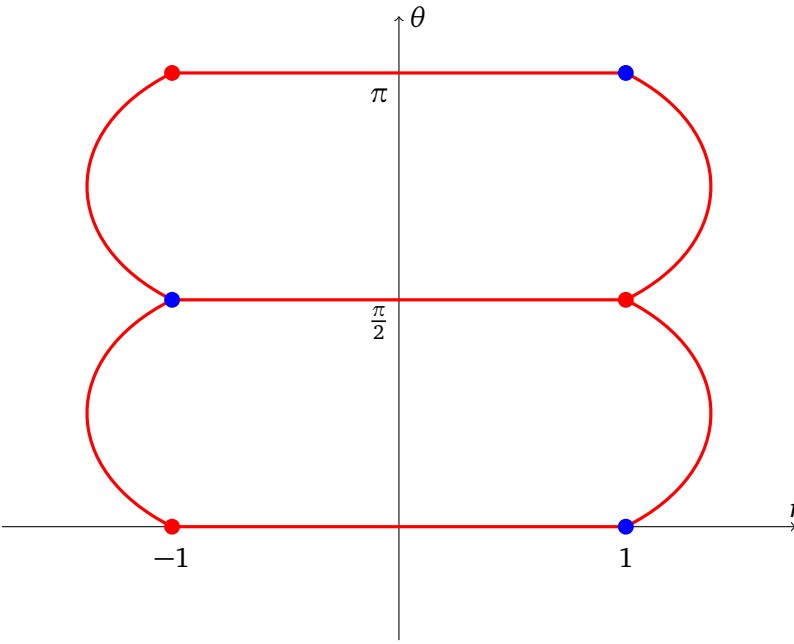

Figure 3: Phase diagram of $H^0$. Red lines correspond to a Luttinger liquid with the Luttinger liquid parameter $K \in (1/2, \infty)$. Red dots correspond to a Luttinger liquid with $K = 1/2$ and blue dots correspond to a quadratic energy-momentum dispersion (and $K = \infty$). Empty regions are magnetically ordered. The phase diagram is symmetric under the reflection $\theta \to -\theta$. The curves on the two sides are $|r| = \sqrt{1 + \sin 2\theta}$ ($0 \leq \theta \leq \pi/2$) and $|r| = \sqrt{1 - \sin 2\theta}$ ($\pi/2 \leq \theta \leq \pi$) with $|r| \geq 1$.

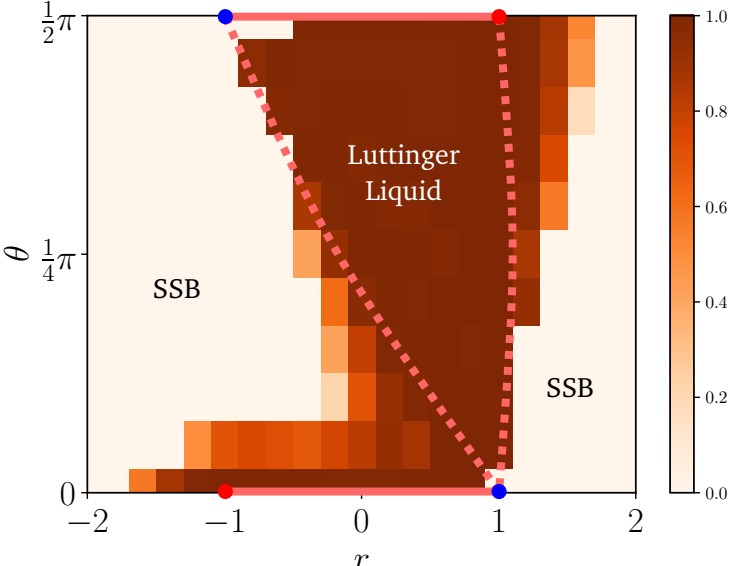

Figure 4: Color plot of central charge $c$ extracted from entanglement entropy of the ground state of $H^1$, calculated by DMRG with system size up to $L = 80$. The dashed lines are conjectured phase boundaries which we cannot determine precisely due to finite size effects. Both the $\theta = 0$ and $\theta = \pi/2$ lines are equivalent to the XXZ model, but are mirror reflection of each other. The red dots are Luttinger liquids with Luttinger liquid parameter $K = 1/2$ (equivalent to $SU(2)_1$ CFT) and the blue dots are gapless states with quadratic dispersion. The phase diagram is symmetric under $\theta \to -\theta$ and $\theta \to \theta + \pi$. "SSB" stands for spontaneous symmetry breaking.

We are not able to solve $H^1$ analytically. We have performed a density matrix renormalization group (DMRG) study and computed the entanglement entropy of the ground state. The extracted central charge $c$ in the $(r, \theta)$ plane are shown in Fig. 4. We briefly describe the phase diagram mapped out from the value of $c$ (additional numerical results are presented in Sec. 3.5). The key feature is that there exists an *extended* region of gapless phase in the phase diagram. The gapless states are Luttinger liquids with a varying Luttinger liquid parameter $K$. In comparison, the gapless region in the phase diagram of $H^0$ has a co-dimension 1. That means, there is one symmetric relevant direction under renormalization group flow for the gapless region of $H^0$, while there is none symmetric relevant direction for the gapless region of $H^1$. This distinction is a consequence of the anomalous $\mathbb{Z}_2$ symmetry of $H^1$. All other regions break the $\mathbb{Z}_2$ symmetry spontaneously, in agreement with the expectation that no symmetric and gapped phase is supported by an anomalous $\mathbb{Z}_2$ symmetry. Comments on a few special lines are in order. (1) On the $\theta = \pi/2$ line (i.e. $w_1^0 = 0$), $H^1$ is equal to $H^0$, so it is equivalent to the XXZ model. It is a Luttinger liquid when $|r| < 1$. (2) On the $r$ axis ($\theta = 0$), $H^1$ is also equivalent to the XXZ model, but it is the mirror image of the $\theta = \pi/2$ line under $r \to -r$. To see that, one may use the Kramers-Wannier duality to map (24) to the XYZ model and find that one of the three parameters $J_x, J_y, J_z$ differs by a minus sign compared to $\theta = \pi/2$. (3) For $= 0$ and $\theta \in [\pi/4, \pi/2]$, it was numerically studied in Ref. [37]. I was found to be a Luttinger liquid, with the Luttinger liquid parameter $K$ varying from 1 to 1/2 as $\theta$ decreases.

### 3.3 Ising fusion category

The simplest example beyond the above two limits is that $G = \mathbb{Z}_2$ and $\mathcal{C}_G = \mathcal{C}_{\text{Ising}}$ the Ising fusion category. The Ising fusion category contains three simple objects, $\mathbb{1}$, $\psi$ and $\sigma$. The nontrivial fusion rules are $\sigma \times \sigma = \mathbb{1} + \psi$, $\psi \times \psi = \mathbb{1}$ and $\psi \times \sigma = \sigma$. Quantum dimensions are $d_{\mathbb{1}} = d_\psi = 1$ and $d_\sigma = \sqrt{2}$. Let $G = \mathbb{Z}_2 = \{0, 1\}$ with group multiplication being addition modulo 2. The Ising category $\mathcal{C}_{\text{Ising}}$ has the following $\mathbb{Z}_2$ grading structure

$$\mathcal{C}_0 = \{\mathbb{1}, \psi\}, \quad \mathcal{C}_1 = \{\sigma\}. \tag{25}$$

Under certain gauge choice, the nontrivial $F$ symbols are given by [42]

$$(F_\sigma^{\psi\sigma\psi})_\sigma = (F_\psi^{\sigma\psi\sigma})_\sigma = -1,$$
$$F_\sigma^{\sigma\sigma\sigma} = \frac{\kappa}{\sqrt{2}} \begin{pmatrix} 1 & 1 \\ 1 & -1 \end{pmatrix}, \tag{26}$$

where $\kappa = \pm 1$ is the Frobenius-Shur indicator distinguishing two variants of Ising fusion category. All other $F$ symbols are equal to 1. The two Ising fusion categories with $\kappa = \pm 1$ can be understood as differing by a nontrivial 3-cocycle in $H^3(\mathbb{Z}_2, U(1)) = \mathbb{Z}_2$. With $\mathcal{C}_{\text{Ising}}$ as the input, we find that our model coincides with that of Ref. [38]. This model can be properly interpreted as the edge model of 2+1D $\mathbb{Z}_2 \times \mathbb{Z}_2^f$ topological superconductors (fermionic SPT phases).

Let us discuss some details of the model for $\mathcal{C}_{\text{Ising}}$. First, we pick the slaved domain-wall variables to be $a_{\mathbf{g}=0} = \mathbb{1}$ and $a_{\mathbf{g}=1} = \sigma$.[4] While both $\{\alpha_i\}$ and $\{x_i\}$ are dynamical variables, the fusion-channel variables $\{x_i\}$ are enough to uniquely label a state. Therefore, we take the short-hand notation

$$|x_{i-1} x_i x_{i+1}\rangle \equiv \left| \begin{array}{c} \quad a_i \quad\quad a_{i+1} \\ \alpha_{i-1} \left| \begin{array}{c} \\ \alpha_i \end{array} \right| \alpha_{i+1} \\ \overline{x_{i-1} \quad x_i \quad x_{i+1}} \end{array} \right\rangle. \tag{27}$$

With the $F$ symbols in (26), the Hamiltonian in (11) reads

$$H_i|\mu\mu\mu\rangle = w_0^{\mathbb{1}}|\mu\mu\mu\rangle + w_1^{\mathbb{1}}|\mu\sigma\mu\rangle,$$
$$H_i|\mu\mu\sigma\rangle = w_0^\sigma|\mu\mu\sigma\rangle + w_1^\sigma|\mu\sigma\sigma\rangle,$$
$$H_i|\mu\sigma\nu\rangle = w_0^{\mu\times\nu}|\mu\sigma\nu\rangle + \delta_{\mu\nu} w_1^{\mathbb{1}}|\mu\mu\mu\rangle,$$
$$H_i|\sigma\mu\mu\rangle = w_0^\sigma|\sigma\mu\mu\rangle + w_1^\sigma|\sigma\sigma\mu\rangle,$$
$$H_i|\mu\sigma\sigma\rangle = w_0^\sigma|\mu\sigma\sigma\rangle + w_1^\sigma|\mu\mu\sigma\rangle,$$
$$H_i|\sigma\mu\sigma\rangle = \sum_\nu \frac{1}{2}\left[w_0^{\mathbb{1}} + (2\delta_{\mu\nu} - 1)w_0^\psi\right]|\sigma\nu\sigma\rangle + \frac{\kappa w_1^{\mathbb{1}}}{\sqrt{2}}|\sigma\sigma\sigma\rangle,$$
$$H_i|\sigma\sigma\mu\rangle = w_0^\sigma|\sigma\sigma\mu\rangle + w_1^\sigma|\sigma\mu\mu\rangle,$$
$$H_i|\sigma\sigma\sigma\rangle = w_0^{\mathbb{1}}|\sigma\sigma\sigma\rangle + \frac{\kappa w_1^{\mathbb{1}}}{\sqrt{2}}(|\sigma\mathbb{1}\sigma\rangle + |\sigma\psi\sigma\rangle), \tag{28}$$

where $\mu, \nu = \mathbb{1}$ or $\psi$. There are five real parameters in this model, $w_0^{\mathbb{1}}$, $w_0^\psi$, $w_0^\sigma$, $w_1^{\mathbb{1}}$ and $w_1^\sigma$ (only three of them are important, while the other two set the zero energy and energy unit, respectively). When $w_0^{\mathbb{1}} = w_0^\psi = w_0^\sigma = 0$, our model reduces exactly to the model of Ref. [38].

---

[4]A different choice is $a_{\mathbf{g}=0} = \psi$ and $a_{\mathbf{g}=1} = \sigma$.

Let us simplify the model by assuming $w_0^{\mathbb{1}} = w_0^{\psi} \equiv w_0$. We further perform an energy shift $H \to H + w_0 \hat{\mathbb{1}}$ and a rescaling $H \to H/\Delta$, with $\Delta = \sqrt{(w_1^{\mathbb{1}})^2 + (w_1^{\sigma})^2}$. Let

$$r = \frac{w_0^{\sigma} - w_0}{\Delta}, \qquad w_1^{\mathbb{1}} = \Delta \cos\theta, \qquad w_1^{\sigma} = \Delta \sin\theta. \tag{29}$$

Then, the Hamiltonian reads

$$
\begin{aligned}
H_i |\mu\mu\mu\rangle &= \cos\theta |\mu\sigma\mu\rangle, \\
H_i |\mu\mu\sigma\rangle &= r|\mu\mu\sigma\rangle + \sin\theta |\mu\sigma\sigma\rangle, \\
H_i |\mu\sigma\nu\rangle &= \delta_{\mu\nu} \cos\theta |\mu\mu\mu\rangle, \\
H_i |\sigma\mu\mu\rangle &= r|\sigma\mu\mu\rangle + \sin\theta |\sigma\sigma\mu\rangle, \\
H_i |\mu\sigma\sigma\rangle &= r|\mu\sigma\sigma\rangle + \sin\theta |\mu\mu\sigma\rangle, \\
H_i |\sigma\mu\sigma\rangle &= \frac{\kappa\cos\theta}{\sqrt{2}} |\sigma\sigma\sigma\rangle, \\
H_i |\sigma\sigma\mu\rangle &= r|\sigma\sigma\mu\rangle + \sin\theta |\sigma\mu\mu\rangle, \\
H_i |\sigma\sigma\sigma\rangle &= \frac{\kappa\cos\theta}{\sqrt{2}} (|\sigma\mathbb{1}\sigma\rangle + |\sigma\psi\sigma\rangle).
\end{aligned}
\tag{30}
$$

There are two continuous parameters $r$ and $\theta$. We will leave the complete phase diagram for future study. At the special point $r = 0$ and $\theta = \frac{\pi}{4}$, we show numerically in Sec. 3.5 that the ground state is the Ising CFT, in agreement with Ref. [38].

Let us discuss the category symmetry in this example. The symmetry operator (6) for $y_{\mathbf{h}} = \sigma$ reads

$$\langle x_1', ..., x_L' | U(\sigma) | x_1, ..., x_L \rangle = \prod_{i=1}^{L} (F_{x_{i+1}'}^{\sigma, x_i, a_{i+1}})_{x_{i+1}}^{x_i'}. \tag{31}$$

Since we take $a_{\mathbf{g}=0} = \mathbb{1}$, a valid state is always of the form

$$|\ldots \sigma\sigma\mu_k\mu_k\mu_k\sigma\sigma\sigma\mu_{k+1}\mu_{k+1}\mu_{k+1}\sigma\sigma\ldots\rangle, \tag{32}$$

i.e., with segments of $\sigma$'s separated by segments of $\mu$'s. The length of each segment can vary. Due to periodic boundary conditions, the number of $\sigma$ segments is always equal to the number of $\mu$ segments. Under the action of $U(\sigma)$, the state in (32) will be mapped to

$$|\ldots \mu_{k-1}'\mu_{k-1}' \,\sigma\sigma\sigma\mu_k'\mu_k'\mu_k'\mu_k'\sigma\sigma\sigma\mu_{k+1}'\mu_{k+1}'\ldots\rangle. \tag{33}$$

With the $F$ symbols in (26), the symmetry operator (31) can be simplified to

$$\langle \{\mu_k'\} | U(\sigma) | \{\mu_k\} \rangle = \left(\frac{\kappa}{\sqrt{2}}\right)^n \prod_{k=1}^{n} (-1)^{(\mu_k + \mu_{k-1})\mu_k'}, \tag{34}$$

where $\mu_k = 0, 1$ corresponds to $\mathbb{1}$ and $\psi$ respectively, and $n$ is the number of $\sigma$ (or $\mu$) segments. Furthermore, one can explicitly check that

$$U(\sigma)^2 = U(\mathbb{1}) + U(\psi), \tag{35}$$

which is consistent with the fusion rule $\sigma \times \sigma = \mathbb{1} + \psi$. Under $U(\psi)$, the state $|\{\mu_k\}\rangle$ is mapped to $|\{\bar{\mu}_k\}\rangle$, with $\bar{\mu}_k = 1 - \mu_k$. We note that the $U(\sigma)$ operator is related to $U_{11}$ in Ref. [38] by $U_{11} = U(\sigma)/\sqrt{2}$. The factor $1/\sqrt{2}$ is important to make $U_{11}$ a *unitary* operator if one restricts to the $U(\psi)$ symmetric subspace (the restriction is necessary when one gauges the $U(\psi)$ symmetry, which is indeed done in Ref. [38]). Note that $U(\sigma)$ is not unitary, justifying that it is a symmetry beyond the description of group.

### 3.4 Tambara-Yamagami category

Tambara-Yamagami category $\mathcal{C}_{\text{TY}}$ is a family of $\mathbb{Z}_2$-graded fusion categories [47]. It is parameterized by a triplet $(A, \chi, \kappa)$, where $A$ is an Abelian group, $\chi$ is a symmetric non-degenerate bicharacter $\chi : A \times A \to U(1)$, and $\kappa = \pm 1$. The simple objects of $\mathcal{C}_{\text{TY}}$ include the elements of $A$ and an object $\sigma$ of quantum dimension $\sqrt{|A|}$, where $|A|$ is the order of $A$. The $\mathbb{Z}_2$-grading structure is given by

$$\mathcal{C}_0 = \{a | a \in A\}, \quad \mathcal{C}_1 = \{\sigma\}. \tag{36}$$

Fusion rules of simple objects in $\mathcal{C}_0$ are given by the group multiplication of $A$. Other fusion rules are $a \times \sigma = \sigma \times a = \sigma$ for any $a \in A$, and $\sigma \times \sigma = \sum_{a \in A} a$. The nontrivial $F$ symbols are given by

$$\begin{aligned}
\left(F_\sigma^{a\sigma b}\right)_\sigma^\sigma &= \left(F_b^{\sigma a\sigma}\right)_\sigma^\sigma = \chi(a, b), \\
\left(F_\sigma^{\sigma\sigma\sigma}\right)_a^b &= \frac{\kappa}{\sqrt{|A|}} \chi^*(a, b),
\end{aligned} \tag{37}$$

where $\kappa$ is the Frobenius-Shur indicator of $\sigma$. If we take $A = \mathbb{Z}_N = \{\mathbb{1}, e, e^2, \ldots, e^{N-1}\}$ with $e^N = \mathbb{1}$, the bicharacter $\chi$ can be explicitly written as

$$\chi(e^m, e^n) = e^{\frac{i2\pi q m n}{N}}. \tag{38}$$

The integer $q$ is coprime with $N$ such that $\chi$ is non-degenerate. For $A = \mathbb{Z}_2$ and $q = 1$, we see that $\mathcal{C}_{\text{TY}}$ becomes $\mathcal{C}_{\text{Ising}}$.

To construct the model out of $\mathcal{C}_{\text{TY}}$, we take the domain wall variables to be $a_{\mathbf{g}=0} = \mathbb{1}$ and $a_{\mathbf{g}=1} = \sigma$. Using the same short-hand notation as Eq. (27), the Hamiltonian is given by

$$\begin{aligned}
H_i |\mu\mu\mu\rangle &= w_0^{\mathbb{1}} |\mu\mu\mu\rangle + w_1^{\mathbb{1}} |\mu\sigma\mu\rangle, \\
H_i |\mu\mu\sigma\rangle &= w_0^\sigma |\mu\mu\sigma\rangle + w_1^\sigma |\mu\sigma\sigma\rangle, \\
H_i |\mu\sigma\nu\rangle &= w_0^{\bar{\mu}\times\nu} |\mu\sigma\nu\rangle + \delta_{\mu\nu} w_1^{\mathbb{1}} |\mu\mu\mu\rangle, \\
H_i |\sigma\mu\mu\rangle &= w_0^\sigma |\sigma\mu\mu\rangle + w_1^\sigma |\sigma\sigma\mu\rangle, \\
H_i |\mu\sigma\sigma\rangle &= w_0^\sigma |\mu\sigma\sigma\rangle + w_1^\sigma |\mu\mu\sigma\rangle, \\
H_i |\sigma\mu\sigma\rangle &= \sum_{\nu,z \in A} \frac{\chi(z, \bar{\mu}\times\nu)}{|A|} w_0^z |\sigma\nu\sigma\rangle + \frac{\kappa w_1^{\mathbb{1}}}{\sqrt{|A|}} |\sigma\sigma\sigma\rangle, \\
H_i |\sigma\sigma\mu\rangle &= w_0^\sigma |\sigma\sigma\mu\rangle + w_1^\sigma |\sigma\mu\mu\rangle, \\
H_i |\sigma\sigma\sigma\rangle &= w_0^{\mathbb{1}} |\sigma\sigma\sigma\rangle + \frac{\kappa w_1^{\mathbb{1}}}{\sqrt{|A|}} \sum_{\mu \in A} |\sigma\mu\sigma\rangle,
\end{aligned} \tag{39}$$

where $\mu, \nu \in A$, and $\bar{\mu}$ is the dual of $\mu$ satisfying $\mu \times \bar{\mu} = \mathbb{1}$.

The bicharacter $\chi$ appears only in the sixth line of Eq. (39). To make a simplification, we take $w_0^x = w_0$ for all $x \in A$. Then, $\sum_{z \in A} \chi(z, \bar{\mu}\times\nu) w_0^z / |A| = \delta_{\mu,\nu} w_0$, which simplifies the sixth line, and the model becomes independent of $\chi$. In addition, we will make an energy shift $H \to H + w_0 \hat{\mathbb{1}}$ and further rescale the Hamiltonian $H \to H/\Delta$, with $\Delta = \sqrt{(w_1^{\mathbb{1}})^2 + (w_1^\sigma)^2}$.

With the same parameterization as (29), the shifted and rescaled Hamiltonian reads

$$
\begin{aligned}
H_i|\mu\mu\mu\rangle &= \cos\theta|\mu\sigma\mu\rangle\,,\\
H_i|\mu\mu\sigma\rangle &= r|\mu\mu\sigma\rangle + \sin\theta|\mu\sigma\sigma\rangle\,,\\
H_i|\mu\sigma\nu\rangle &= \delta_{\mu\nu}\cos\theta|\mu\mu\mu\rangle\,,\\
H_i|\sigma\mu\mu\rangle &= r|\sigma\mu\mu\rangle + \sin\theta|\sigma\sigma\mu\rangle\,,\\
H_i|\mu\sigma\sigma\rangle &= r|\mu\sigma\sigma\rangle + \sin\theta|\mu\mu\sigma\rangle\,,\\
H_i|\sigma\mu\sigma\rangle &= \frac{\kappa\cos\theta}{\sqrt{|A|}}|\sigma\sigma\sigma\rangle\,,\\
H_i|\sigma\sigma\mu\rangle &= r|\sigma\sigma\mu\rangle + \sin\theta|\sigma\mu\mu\rangle\,,\\
H_i|\sigma\sigma\sigma\rangle &= \frac{\kappa\cos\theta}{\sqrt{|A|}}\sum_{\mu\in A}|\sigma\mu\sigma\rangle\,.
\end{aligned}
\tag{40}
$$

For $A = \mathbb{Z}_2$, it reduces to Eq. (30) of the Ising fusion category.

## 3.5 Numerical results

In this section, we present some numerical results on the models introduced in Sec. 3.2.1, 3.3 and 3.4. We compute the energy spectrum by exact diagonalization (ED) and entanglement entropy of the ground state obtained by density matrix renormalization group (DMRG) [48].

Our main interests are the gapless states described by conformal field theory (CFT). According to CFT, the low-lying energies of a system of finite size $L$ in periodic boundary conditions take the form [49]

$$
E = E_1 L + \frac{2\pi v}{L}\left(-\frac{c}{12} + h + \bar{h}\right)\,,
\tag{41}
$$

where the velocity $v$ is an overall scale factor and $c$ is the central charge of the CFT. The scaling dimensions $h + \bar{h}$ take the form $h = h^0 + n$, $\bar{h} = \bar{h}^0 + \bar{n}$, with $n$ and $\bar{n}$ non-negative integers, and $h^0$ and $\bar{h}^0$ are the holomorphic and antiholomorphic conformal weights of the primary fields in the given CFT. We will compare the ED spectrum to Eq. (41). Instead of using (41), we compute the central charge $c$ from the entanglement entropy $S$ of the many-body ground state. Under periodic boundary conditions, it is given by [50]

$$
S(x) = \frac{c}{3}\ln\left(\frac{L}{\pi}\sin\left(\frac{\pi x}{L}\right)\right) + a\,,
\tag{42}
$$

where $L$ is the system size, $x$ is the length of the subsystem used to calculate the entanglement entropy, and $a$ is a non-universal constant. For computation of $S(x)$, we use DMRG to access larger system sizes. We use the ITensor package for DMRG calculations. [51]

Below we present the results for the $\mathbb{Z}_2$ SPT edge models $H^0$ (16) and $H^1$ (17), Ising fusion category model (30), and Tambara-Yamagami category model (40) with $A = \mathbb{Z}_3$. We remark that the Ising category model is the same as Tambara-Yamagami model with $A = \mathbb{Z}_2$. Also, the $\mathbb{Z}_2$ edge models $H^0$ and $H^1$ are equivalent to the Tambara-Yamagami models with $A = \mathbb{Z}_1$, for $\kappa = 1$ and $\kappa = -1$ respectively. Therefore, we put the numerical results together and make a comparison. We will leave a complete study of the phase diagrams for future study. In this work, we mainly focus on

$$
w_{\mathbf{g}}^z = 1\,,\quad \forall z\,,\mathbf{g}\,,
\tag{43}
$$

i.e., $r = 0$ and $\theta = \pi/4$ in (24), (30), and (40). These values are chosen without any priori knowledge, but only because of simplicity. It turns out that all models with $\kappa = 1$ are CFTs at parameters in (43), while the cases with $\kappa = -1$ are less conclusive. We remark that the gapless state at the parameters (43) for the $\kappa = 1$ Ising and $\mathbb{Z}_3$ Tambara-Yamagami models

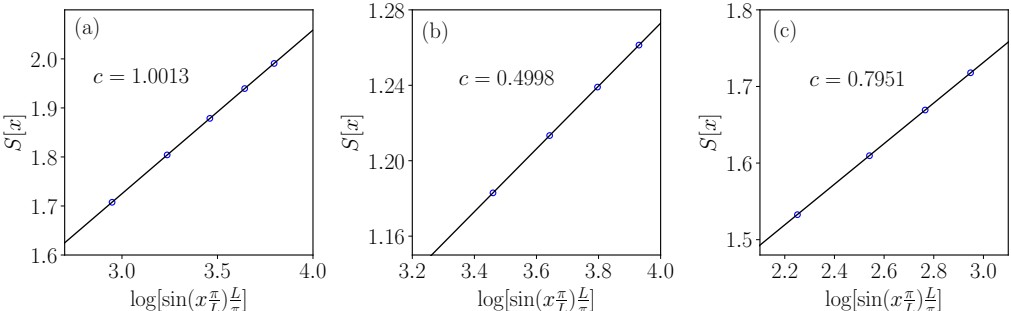

Figure 5: Entanglement entropy $S$ of different models: (a) $\mathbb{Z}_2$ edge model $H^0$ with system size $L = 60, 80, 100, 120, 140$; (b) Ising category model with $L = 100, 120, 140, 160$; and (c) TY category model with $L = 30, 40, 50, 60$. All results are obtained with periodic boundary conditions and subsystem size $x = L/2$. For both Ising category and $\mathbb{Z}_3$ Tambara-Yamagami category models, the parameters $\kappa = 1$, $r = 0$ and $\theta = \pi/4$. For the $\mathbb{Z}_2$ edge model $H^0$, parameters are set at $\omega_0^0 - \omega_0^1 = 2$ and $\omega_1^0 = \omega_1^1 = 1$.

is *not* an isolated point in the parameter space. Our preliminary calculations show that there exists an extended nearby gapless region, similar to Fig. 4, which we will present elsewhere after a more careful numerical investigation.

### 3.5.1 $\kappa = 1$

Let us first consider the models with $\kappa = 1$ and the parameters in (43). With the ground state from DMRG calculations, we obtain the entanglement entropy $S$ and fit the results according to Eq. (42), as shown in Figs. 5. For the Ising and Tambara-Yamagami category models, we find $c \approx 1/2$ and $c \approx 4/5$, respectively. This indicates that, with the parameters (43), the two models belong to the critical Ising and critical 3-state Potts universality classes, respectively. This is verified by computing the low-energy ED spectra, which fit well with the CFT prediction Eq. (41) (see Fig. 6). (Another $c = 4/5$ CFT is the tetra-critical Ising theory, but its low-energy spectrum does not fit into our ED spectrum.)

The model $H^0$ in (16) can be solved exactly by mapping to XYZ model. Nevertheless, we did some numerical calculations for verification. It is gapped for the parameters in (43), so instead we set $w_0^0 - w_0^1 = 2$ and $w_1^0 = w_1^1 = 1$ (it is equivalent to $r = -\sqrt{2}$ and $\theta = \pi/4$ in the Tambara-Yamagami Hamiltonian (40) with $A = \mathbb{Z}_1$). With this setting, the low-energy physics is described by double copies of the Ising CFT, see Fig. 6. It is equivalent to a free massless complex fermion after a $\mathbb{Z}_2$ orbifolding [52], which is a $K = 1$ Luttinger liquid. This agrees with the analytic results [46].

### 3.5.2 $\kappa = -1$

For $\kappa = -1$, all models display a much stronger finite-size effect than the case of $\kappa = 1$. So far, we have only done a relatively complete search of gapless regions for the $\mathbb{Z}_2$ edge model $H^1$. The phase diagram mapped out from the central charge is shown in Fig. 4 (see discussions in Sec. 3.2.1). In all the gapless regions, we find the central charge $c = 1$, i.e., a Luttinger liquid. At the parameters $r = 0$ and $\theta = \pi/4$, the numerical results of entanglement entropy are shown Fig. 7(a). The ED spectrum at $L = 16$ is also shown in Fig. 7(b), but not much information can be extracted due to strong finite size effect.

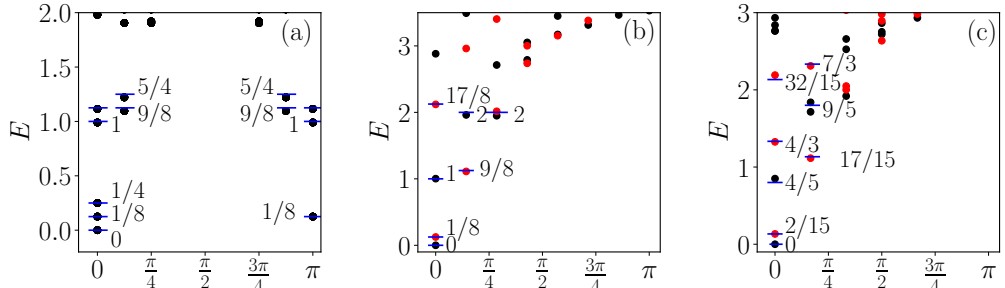

Figure 6: Finite-size energy spectra of (a) $\mathbb{Z}_2$ edge model $H^0$ at $L = 16$, (b) Ising category model $L = 14$ and (c) $\mathbb{Z}_3$ TY category model at $L = 12$, corresponding to double Ising CFT, Ising CFT and critical 3-state Potts CFT, respectively. Dots are numerical results and bars are analytic predictions [49]. Parameters are same as in Fig. 5 and energies are properly shifted and rescaled. All dots in (a) and (b) are non-degenerate. Black and red dots in (b) correspond to the eigenvalue $+1$ and $-1$ of $U(\psi)$, respectively. Every red dot in (c) is doubly degenerate, corresponding to the eigenvalue $U(e) = e^{\pm i2\pi/3}$ respectively, with $e$ being the generator of $\mathbb{Z}_3$.

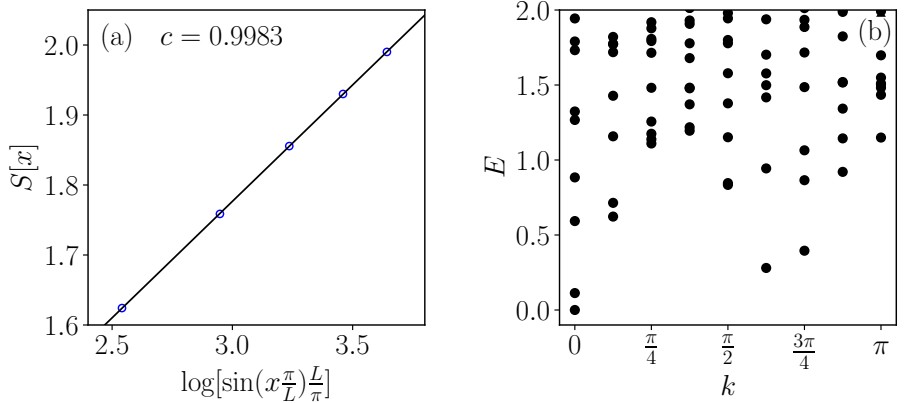

Figure 7: (a) Entanglement entropy $S(x)$ at $L = 40, 60, 80, 100, 120$ and (b) energy spectrum for $H^1$ at $L = 16$. Parameters are $r = 0$ and $\theta = \pi/4$.

For Ising category and $\mathbb{Z}_3$ Tambara-Yamagami category, we find that models are gapped at $r = 0$ and $\theta = \pi/4$, as we observe $S$ decreases to a constant as $L$ increases (not shown here). We have searched for gapless spectra at other values of parameters and found some evidence. Nevertheless, it is not conclusive yet. We leave a careful numerical investigation for the future.

## 3.6  $SU(2)_k$ theory

Another family of $\mathbb{Z}_2$-graded category is associated with anyons from $SU(2)_k$ theory. We denote the category as $\mathcal{C}_{SU(2)_k}$. The objects in $\mathcal{C}_{SU(2)_k}$ are closely related to the ordinary $SU(2)$ spins, which can be labeled by $s = 0, \frac{1}{2}, 1, ..., \frac{k}{2}$, with $k$ being a positive integer. There are $k+1$ objects in total. The fusion rule between $s$ and $s'$ is given by

$$s \times s' = \sum_{s''=|s-s'|}^{\min(s+s', k-s-s')} s'', \tag{44}$$

where the summation is incremented by 1, similar to addition of ordinary angular momenta. One can see that integer spins are closed under fusion. By taking $\mathcal{C}_0 = \{0, 1, ...\}$

and $\mathcal{C}_1 = \{\frac{1}{2}, \frac{3}{2}, ...\}$, we have the following decomposition

$$\mathcal{C}_{SU(2)_k} = \mathcal{C}_0 \oplus \mathcal{C}_1. \tag{45}$$

This gives the $\mathbb{Z}_2$-grading structure of $\mathcal{C}_{SU(2)_k}$: $\mathcal{C}_0$ is closed under fusion, two objects from $\mathcal{C}_1$ fuse into objects in $\mathcal{C}_0$, and fusing an object from $\mathcal{C}_0$ and an object from $\mathcal{C}_1$ gives objects in $\mathcal{C}_1$. To build our model (11), we need the $F$ symbols in $\mathcal{C}_{SU(2)_k}$. The $F$ symbols are known explicitly [53] (see also, e.g., Ref. [30]), but we will not list them here. It is interesting to perform a detailed numerical study of this family of models in the future.

We give a brief further discussion on the $k = 3$ case. It is closely related to the famous Fibonacci anyon. In this case, $\mathcal{C}_{SU(2)_3} = \{0, 1\} \oplus \{\frac{1}{2}, \frac{3}{2}\}$. The quantum dimensions are $d_0 = d_{\frac{3}{2}} = 1$ and $d_1 = d_{\frac{1}{2}} = \frac{\sqrt{5}+1}{2}$. The object $s = 1$ corresponds the Fibonacci anyon. Therefore, $\mathcal{C}_0 = \{0, 1\}$ is the usual Fibonacci category, and $\mathcal{C}_{SU(2)_3}$ is a $\mathbb{Z}_2$ extension of $\mathcal{C}_0$. (There are two kinds of $\mathbb{Z}_2$ extensions of the Fibonacci category, whose $F$ symbols differ by the non-trivial 3-cocycle in $H^2(\mathbb{Z}_2, U(1))$, see Eq. (4).) It is interesting to study the low-energy physics of our model (11) based on $\mathcal{C}_{SU(2)_3}$, and compare it to the golden chain model [29] whose low-energy physics is captured by the tricritical Ising conformal field theory.

## 3.7 $\mathcal{C}_G$ from groups

From group extensions, one can define many $G$-graded unitary fusion categories. Consider the short exact sequence

$$1 \to N \to \mathcal{C}_G \to G \to 1, \tag{46}$$

where $N$ and $G$ are two finite groups, and $\mathcal{C}_G$ is called an extension of $G$ by $N$. The group $N$ is a normal subgroup of $\mathcal{C}_G$ and $G$ is isomorphic to the quotient group $\mathcal{C}_G/N$. Let $\mathcal{C}_0 \equiv N$, and $\mathcal{C}_{\mathbf{g}} \equiv \mathbf{g}N$ to be the coset in $\mathcal{C}_G$ associated with $\mathbf{g} \in G$. Then, $\mathcal{C}_G$ has the following decomposition

$$\mathcal{C}_G = \bigoplus_{\mathbf{g} \in G} \mathbf{g}N = \bigoplus_{\mathbf{g} \in G} \mathcal{C}_{\mathbf{g}}. \tag{47}$$

Taking a 3-cocycle $\nu_3 \in \mathcal{Z}^3(\mathcal{C}_G, U(1))$, we can then regard the doublet $(\mathcal{C}_G, \nu_3)$ as a $G$-graded fusion category. Without causing confusion, we will sometimes simply call $\mathcal{C}_G$ the $G$-graded fusion category, although it is only a group in this subsection.

Given $N$ and $G$, the extended group $\mathcal{C}_G$ is not unique. Let $a, b, ...$ be elements of $N$, and $\mathbf{g}, \mathbf{h}, ...$ be elements of $G$. Then, group elements in $\mathcal{C}_G$ can be labeled by $a_{\mathbf{g}}$, with $a$ running through elements in $N$ and $\mathbf{g}$ running through elements in $G$.[5] To specify the multiplication law of $\mathcal{C}_G$, we need two pieces of data: (i) a group homomorphism $\rho : G \to \text{Out}(N)$, where $\text{Out}(N)$ is the outer automorphism group of $N$, and (ii) a torsor $\mu$ in $H^2_\rho(G, Z(N))$, where $Z(N)$ is the center of $N$. Let $\mathbf{g} \in G$ and $\rho_{\mathbf{g}} \equiv \rho(\mathbf{g}) \in \text{Out}(N)$. Then, $\rho_{\mathbf{g}}(a)$ describes the action of $\mathbf{g}$ on $a \in N$. The torsor $\mu$ is a function $\mu : G \times G \to Z(N)$, which satisfies the twisted 2-cocycle conditions associated with $\rho$. Given $\rho$ and $\mu$, group multiplication in $\mathcal{C}_G$ can be defined by

$$a_{\mathbf{g}} \times b_{\mathbf{h}} = [a \cdot \rho_{\mathbf{g}}(b) \cdot \mu(\mathbf{g}, \mathbf{h})]_{\mathbf{gh}}, \tag{48}$$

where "·" denotes group multiplication in $N$. It is clear that the group multiplication respects the $G$-grading structure.

A cocycle $\nu_3$ in $\mathcal{Z}^3(\mathcal{C}_G, U(1))$ can also be parameterized by a set of data associated with $N$ and $G$. Based on the Lyndon-Hochschild-Serre spectral sequence, it was shown in Ref. [54] that

---

[5]This notation has a different meaning from $a_{\mathbf{g}}$ elsewhere in this paper. In this subsection, $a \in N$ and $\mathbf{g} \in G$ are independent. In other parts of the paper, $a \in \mathcal{C}_G$ and $\mathbf{g}$ denotes the grading property of $a$.

$v_3$ valued at general $a_\mathbf{g}, b_\mathbf{h}, c_\mathbf{k}$ can be fully determined by $v_3$ at special elements of $\mathcal{C}_G$. Specifically, $v_3(a_\mathbf{g}, b_\mathbf{k}, c_\mathbf{k})$ is determined by $v_3(a, b, c)$, $v_3(a, b, 1_\mathbf{g})$, $v_3(a, 1_\mathbf{g}, 1_\mathbf{h})$ and $v_3(1_\mathbf{g}, 1_\mathbf{h}, 1_\mathbf{k})$, with $a, b, c \in N$ and $\mathbf{g}, \mathbf{h}, \mathbf{k} \in G$ (note that $a \equiv a_\mathbf{1}$). We refer the readers to Ref. [54] for the general parameterization. Here, we only consider the special case that both $\rho$ and $\mu$ are trivial. In this case, $\mathcal{C}_G = N \times G$, and the 3-cocycle $v_3$ is simply the product of the four special pieces

$$v_3(a_\mathbf{g}, b_\mathbf{h}, c_\mathbf{k}) = v_3(a, b, c)\, v_3(a, b, 1_\mathbf{k})\, v_3(a, 1_\mathbf{h}, 1_\mathbf{k})\, v_3(1_\mathbf{g}, 1_\mathbf{h}, 1_\mathbf{k}). \tag{49}$$

This expression can be well understood from the Künneth formula

$$\mathcal{H}^3(N \times G, U(1)) = \mathcal{H}^3(N, U(1)) \oplus \mathcal{H}^1(G, \mathcal{H}^2(N, U(1)))$$
$$\oplus \mathcal{H}^2(G, \mathcal{H}^2(N, U(1)) \oplus \mathcal{H}^3(G, U(1)). \tag{50}$$

The four pieces in (49) have a one-to-one correspondence to elements in the cohomology groups on the right hand side of (50). The parameterization of $v_3$ with general $\rho$ and $\mu$ is more complicated but follows a similar structure.

With $(\mathcal{C}_G, v_3)$, we can construct a lattice model following Sec. 2. For simplicity, we assume that $\rho$ and $\mu$ are trivial. The domain degrees of freedom $\alpha_i$ take values in $G$, and domain walls $a_i$ and $x_i$ take values in a proper coset $\mathcal{C}_\mathbf{g} = \mathbf{g}N$. To build up the model, we need to manually pick up a fixed element $\bar{b} \in N$ for every $\mathbf{g} \in G$, which together select a representative $\bar{b}_\mathbf{g}$ from each coset $\mathcal{C}_\mathbf{g}$. Then, on the $i$th domain wall, it lives an object $a_i = \bar{b}_{\alpha_{i-1}^{-1}\alpha_i} \equiv (\bar{b}_i)_{\alpha_{i-1}^{-1}\alpha_i}$ (we use $\bar{b}_i$ to denote the $\bar{b} \in N$ that lives on the $i$th domain wall). The fusion channel $x_i \in \mathcal{C}_{\alpha_i}$ and let us denote $x_i \equiv (d_i)_{\alpha_i}$, with $d_i \in N$. With fusion rules, we have $x_i = x_{i-1}a_i$ and $d_i = d_{i-1}\bar{b}_i$. Two features of the Hilbert space deserve to be mentioned. (1) Given $\{\alpha_i\}$ and $\{a_i\}$, there are only $|N|$ possible $\{x_i\}$:

$$x_i = (d_i)_{\alpha_i}, \quad \text{with} \quad d_i = d_0 \prod_{j=1}^{i} \bar{b}_j, \tag{51}$$

where $d_0$ runs though elements in $N$. Hence, states in the Hilbert space can be labeled as $|\{\alpha_i\}, d_0\rangle$. We will give a further discussion on $d_0$ below. (2) The periodic boundary condition requires that

$$\prod_{i=1}^{L} \bar{b}_i = 1. \tag{52}$$

It follows from $d_{L+1} = d_1 \prod_i \bar{b}_i$ and $d_{L+1} = d_1$.

The symmetry operator $U(y_\mathbf{h})$, defined in Eq.(6), is given by

$$\langle\{\mathbf{h}\alpha_i, x_i'\}|U(y_\mathbf{h})|\{\alpha_i, x_i\}\rangle = \prod_{i=1}^{L} v_3^*(y_\mathbf{h}, (d_i)_{\alpha_i}, \bar{b}_{\alpha_i^{-1}\alpha_{i+1}}),$$
$$= \prod_{i=1}^{L} v_3^*(y, d_i, \bar{b}_i)\, v_3^*(y, d_i, 1_{\alpha_i^{-1}\alpha_{i+1}})$$
$$\times v_3^*(y, 1_{\alpha_i}, 1_{\alpha_i^{-1}\alpha_{i+1}})\, v_3^*(1_\mathbf{h}, 1_{\alpha_i}, 1_{\alpha_i^{-1}\alpha_{i+1}}), \tag{53}$$

where we have inserted Eq. (49) into the second equality. Note that the action of $y_\mathbf{h}$ gives $x_i' = y_\mathbf{h} \times x_i = [y \cdot d_i]_{\alpha_i'}$. The Hamiltonian can be written down following Sec. 2.4.

Let us compare this example to that in Sec. 3.2. First, while both examples realize the symmetry $(\mathcal{C}_G, v_3)$, the allocation of degrees of freedom from $N$ and $G$ on the lattice are different. In the example of Sec. 3.2, $\alpha_i$ can fluctuate freely in $\mathcal{C}_G$. In the current example, $\alpha_i$ fluctuates only within $G$, while elements from $N$ which live on the domain walls are constrained. Accordingly, to realize the same symmetry $(\mathcal{C}_G, v_3)$, the current example could have a smaller

Hilbert space as long as one properly divides $\mathcal{C}_G$ into $N$ and $G$. This is useful for numerical investigations. Second, the degree of freedom $d_0$, absent in the example of Sec. 3.2, is a *global* degree of freedom. It enters every $d_i$ and cannot be changed by any local operators. This makes the ground states of local Hamiltonian to be $|N|$-fold degenerate. For simplicity, let us consider the case $G = 1$, i.e., with no $\{\alpha_i\}$ degrees of freedom. In this case, the whole Hilbert space is $|N|$ dimensional, and the Hamiltonian is proportional to the identity matrix. Since the $|N|$-fold degenerate ground-state space transforms non-trivially under $N$, the group $N$ is actually "spontaneously broken". To make the "symmetry breaking" claim more explicit, let us allow $\{\bar{b}_i\}$ to fluctuate (see a more general discussion around Eq. (71)). Let us denote the states in the enlarged Hilbert space as $|\{b_i\}, d_0\rangle$, with the "‾" removed to indicate that they can fluctuate. Note that $\{b_i\}$ are subject to the constraint (52). The state $|\{b_i\}, d_0\rangle$ can be equivalently labeled as $|\{d_i\}\rangle$, with $d_i = d_{i-1}^{-1} b_i$. In the notation $|\{d_i\}\rangle$, each $d_i$ can fluctuate freely in $N$. With this preparation, the selection of $\bar{b}_i$ corresponds to adding the action

$$H' = -\Delta \sum_i \delta_{d_{i-1}^{-1} d_i, \bar{b}_i}, \tag{54}$$

and taking the limit $\Delta \to \infty$. The interaction $H'$ describes a kind of "ferromagnetic" interaction between $d_i$ and $d_{i-1}$, and it is symmetric under $N$. In particular, if $\bar{b}_i = 1$, the interaction becomes $-\delta_{d_{i-1}, d_i}$. It is now obvious that the ground state of $H'$ spontaneously breaks the symmetry group $N$.

## 4 Discussions

### 4.1 Gauge choice of $F$ and 1D SPT states

In category theory, $F$ symbol is not a gauge invariant quantity. Given $\mathcal{C}_G$, one can take different gauge choices for $F$. Since the Hamiltonian (11) explicitly depends the $F$ symbol, we expect the ground states to be dependent on the gauge choices of $F$ too. In fact, gauge-equivalent $F$ symbols can lead to *inequivalent* $\mathcal{C}_G$-symmetric ground states. Loosely speaking, these distinct ground states can be thought of differing by 1D SPT states of $\mathcal{C}_G$ category symmetry.

To demonstrate this point, we consider the special example $\mathcal{C}_G = (G, \nu_3)$, with $\nu_3$ being a trivial 3-cocycle. Recall from Sec. 3.2 that the $F$ symbol is determined by $\nu_3$, and our model can be thought of as an effective edge model of a 2D SPT bulk. When $\nu_3$ is a trivial 3-cocycle, it can be written as

$$\nu_3(\mathbf{g}, \mathbf{h}, \mathbf{k}) = \frac{c_2(\mathbf{h}, \mathbf{k}) c_2(\mathbf{g}, \mathbf{hk})}{c_2(\mathbf{gh}, \mathbf{k}) c_2(\mathbf{g}, \mathbf{h})}, \tag{55}$$

where $c_2$ is an arbitrary 2-cochain, i.e., a function $c_2 : G \times G \to U(1)$. Inserting (55) into the expression (14) of $U(\mathbf{g})$, we have

$$U(\mathbf{g})|\alpha_1, \alpha_2 \ldots, \alpha_L\rangle = \prod_i \frac{c_2(\mathbf{g}\alpha_i, \alpha_i^{-1}\alpha_{i+1})}{c_2(\alpha_i, \alpha_i^{-1}\alpha_{i+1})} |\mathbf{g}\alpha_1, \mathbf{g}\alpha_2, \ldots, \mathbf{g}\alpha_L\rangle, \tag{56}$$

If we take a local unitary transformation to the new basis

$$|\alpha_1, \ldots, \alpha_L\rangle\!\rangle = \prod_i c_2(\alpha_i, \alpha_i^{-1}\alpha_{i+1})|\alpha_1, \ldots, \alpha_L\rangle, \tag{57}$$

the symmetry $U(\mathbf{g})$ acts in the conventional onsite fashion

$$U(\mathbf{g})|\alpha_1, \alpha_2 \ldots, \alpha_L\rangle\!\rangle = |\mathbf{g}\alpha_1, \mathbf{g}\alpha_2, \ldots, \mathbf{g}\alpha_L\rangle\!\rangle. \tag{58}$$

This onsite form can be achieved because $v_3$ is a trivial 3-cocycle, or equivalently because the corresponding 2D SPT bulk is trivial. In the new basis, the Hamiltonian (13) of our model is given by

$$\langle\!\langle \alpha_{i-1}, \alpha_i', \alpha_{i+1}|H_i|\alpha_{i-1}, \alpha_i, \alpha_{i+1}\rangle\!\rangle = w_{\mathbf{h}_i}^{z_i} \frac{c_2(\alpha_{i-1}^{-1}\alpha_i, \alpha_i^{-1}\alpha_{i+1})}{c_2(\alpha_{i-1}^{-1}\alpha_i', \alpha_i'^{-1}\alpha_{i+1})} \, . \tag{59}$$

It is straightforward to see that the Hamiltonian is symmetric under the onsite symmetry (58).

So far, $c_2$ is an arbitrary 2-cochain. If we take $w_{\mathbf{h}_i}^{z_i} = 1$ and $c_2$ to be a 2-cocycle, i.e., $c_2(\mathbf{h}, \mathbf{k})c_2(\mathbf{g}, \mathbf{hk}) = c_2(\mathbf{gh}, \mathbf{k})c_2(\mathbf{g}, \mathbf{h})$, the Hamiltonian (59) can be rewritten as

$$\langle\!\langle \alpha_{i-1}, \alpha_i', \alpha_{i+1}|H_i|\alpha_{i-1}, \alpha_i, \alpha_{i+1}\rangle\!\rangle = \frac{c_2(\alpha_{i-1}^{-1}\alpha_i, \alpha_i^{-1}\alpha_i')}{c_2(\alpha_i^{-1}\alpha_i', \alpha_i'^{-1}\alpha_{i+1})} \, . \tag{60}$$

It is precisely the fixed-point group-cohomology model of 1D SPT states proposed in Ref. [2]. It is known that inequivalent 2-cocycles $c_2$ give rise to topologically distinct gapped SPT states of symmetry group $G$. Therefore, we see that for the trivial $v_3$, different gauge choices (i.e., different $c_2$) give rises to topologically distinct SPT phases. We remark that, in general, $c_2$ is not a 2-cocycle as we do not require our model to sit at a fixed point. Our model may also break symmetry spontaneously.

If $v_3$ is a non-trivial 3-cocycle, we cannot write $v_3$ into the form (55). However, we can still take different gauge choices by shifting $v_3(\mathbf{g}, \mathbf{h}, \mathbf{k}) \to v_3(\mathbf{g}, \mathbf{h}, \mathbf{k}) \frac{c_2(\mathbf{h},\mathbf{k})c_2(\mathbf{g},\mathbf{hk})}{c_2(\mathbf{gh},\mathbf{k})c_2(\mathbf{g},\mathbf{h})}$. A nontrivial $v_3$ means the symmetry group $G$ carries 't Hooft anomaly. The ground state cannot be simultaneously non-degenerate, gapped and symmetric. Let us assume a gapless and symmetric ground state, and discuss a potential implication from different gauge choices of $v_3$. From the above discussion on the trivial $v_3$ case, we speculate that different gauge choices of non-trivial $v_3$ correspond to the gapless state to be stacked with different 1D SPT states of the same group $G$. Since SPT states are gapped, stacking them will not modify the gapless spectrum dramatically. However, topological properties of the gapless system might be modified. We do not know the precise meaning of topological properties of a gapless system yet. It would be interesting to explore this question in the future. We note that it might have a close relation to gapless SPT phases discussed in Refs. [55, 56].

For a general category $\mathcal{C}_G$, it is also possible to study "generalized SPT" phases under appropriate definitions. A reasonable definition is that an SPT state is a gapped, symmetric and non-degenerate ground state of a Hamiltonian that respects the category symmetry $\mathcal{C}_G$. However, SPT state may not always exists. For example, as just discussed, if $\mathcal{C}_G = (G, v_3)$ and $v_3$ is a nontrivial 3-cocycle, it cannot support systems with a gapped symmetric unique ground state. If a category symmetry does not support (trivial or nontrivial) SPT phases, it is called *anomalous*, generalizing the concept of 't Hooft anomaly of group-like symmetries. Criteria on whether a category symmetry is anomalous have been studied in Ref. [27]. For non-anomalous category symmetry, we expect that different gauge choices of $F$ correspond to different $\mathcal{C}_G$-symmetric SPT phases. For anomalous category symmetries, implications of different gauge choices of $F$ is subtler, as the meaning of "stacking" shall be elaborated before we generalize the case of groups. All these are interesting questions to explore in the future.

## 4.2 Relation to boundary of 2+1D topological phases

Our model with symmetry (6) and Hamiltonian (8) can be viewed as a boundary theory of 2+1D topological phases. More precisely, in this subsection, we show that it can be viewed as a boundary theory of 2+1D symmetry enriched string-net model (SESN) defined on a disk geometry under certain choice of boundary conditions.

Let us start with a brief review of the SESN model. It is defined on a trivalent lattice with the orientated links. The input data is a $G$-graded unitary fusion category $\mathcal{C}_G$. There are two types of degrees of freedom on the lattice. On each oriented link, there lives a $|\mathcal{C}_G|$-component "spin". Each component of the spin is a simple object $a \in \mathcal{C}_G$, which is also called a string type. On each plaquette, there lives a $|G|$-component "spin", with each component being a group element $\mathbf{g} \in G$, as see Fig. 8. The basis vectors of the Hilbert space can be denoted as $|\{a_l, \mathbf{g}_p\}\rangle$, with $l$ runs over the links and $p$ runs over the plaquettes. The Hamiltonian is

$$H = -\sum_v A_v - \sum_l P_l - \sum_p B_p, \tag{61}$$

where the sum runs over the vertices ($v$), the links ($l$), and plaquettes ($p$). All $A_v$, $P_l$ and $B_p$ are projector operators, with eigenvalues being 0 and 1. The term $A_v = \delta_{abc}$ when acts on basis vectors, where $a$, $b$ and $c$ are the three strings meeting at vertex $v$, $\delta_{abc} = 1$ if $a, b, c$ satisfy the fusion rules of $\mathcal{C}_G$ and $\delta_{abc} = 0$ otherwise (again, we assume $\mathcal{C}_G$ is fusion multiplicity free). Assuming the string type on link $l$ is $a_{\mathbf{g}} \in \mathcal{C}_G$, the term $P_l = \delta_{\mathbf{g}, \mathbf{g}_p^{-1} \mathbf{g}_q}$, where $\mathbf{g}_p$ and $\mathbf{g}_q$ are the plaquette spins on left and right of the link $l$, respectively (under an appropriate orientation convention). The term $B_p$ is defined as

$$B_p = \frac{1}{D^2} \sum_{s \in \mathcal{C}_G} d_s B_p^s \tilde{U}_p^{\mathbf{g}_s}, \tag{62}$$

where $d_s$ is the quantum dimension of $s$ and $D = \sqrt{\sum_s d_s^2}$ is the total quantum dimension. The notation $\mathbf{g}_s$ is used to denote $s \in \mathcal{C}_{\mathbf{g}_s}$. The term $\tilde{U}_p^{\mathbf{g}_s}$ flips the plaquette spin in the following way

$$\tilde{U}_p^{\mathbf{g}_s} |\mathbf{g}_p\rangle = |\mathbf{g}_p \mathbf{g}_s\rangle, \tag{63}$$

where irrelevant spins are omitted in the notation $|\mathbf{g}_p\rangle$. The $B_p^s$ can be understood as creating a string $s$ inside the plaquette $p$ and fusing it into the boundary strings of the plaquette, so the matrix element of $B_p^s$ is a product of $F$ symbols. A nice property of the SESN model is that all the projectors $A_v$, $P_l$ and $B_p$ commute with each other, making the model exactly solvable. The SESN model has an onsite $G$ symmetry

$$U^{\mathbf{g}} = \prod_p U_p^{\mathbf{g}}, \qquad U_p^{\mathbf{g}} |\mathbf{g}_p\rangle = |\mathbf{g}\mathbf{g}_p\rangle. \tag{64}$$

The SESN model realizes a topological order which mathematically is the Drinfeld center $\mathcal{Z}(\mathcal{C}_0)$. Since it is $G$ symmetric, it is an SET state of the $\mathcal{Z}(\mathcal{C}_0)$ topological order. Readers are referred to Refs. [39, 40] for more details.

Now we consider the 2D SESN model on a disk geometry. In Fig. 8, the orange region represents the string-net bulk while the blue region represents the boundary. We will see that our 1D model lives in the subspace of the 2D SESN model after projecting the bulk into its ground state. To match the notation of our 1D model (Fig. 1), we have labeled the corresponding $\alpha_i$, $a_i$, and $x_i$ in the blue region in Fig. 8: the plaquette spins $\alpha_i \in G$ correspond to the domain variables in the 1D model, and the link spins $a_i \in \mathcal{C}_G$ and $x_i \in \mathcal{C}_G$ correspond to the domain wall variables. For convenience, we will call $\{\alpha_i, a_i, x_i\}$ the *boundary spins* below. Let us consider the following Hamiltonian

$$H_{\text{disk}} = -\sum_{v \in \text{all}} A_v - \sum_{l \in \text{all}} P_l - \sum_{p \in \text{bulk}} B_p, \tag{65}$$

where $p$ runs only over the orange "bulk plaquette" in Fig. 8. The projectors $A_v$, $P_l$ and $B_p$ are the same as above. There is an ambiguity on $P_l$ for the outermost links of the disk. To fix this

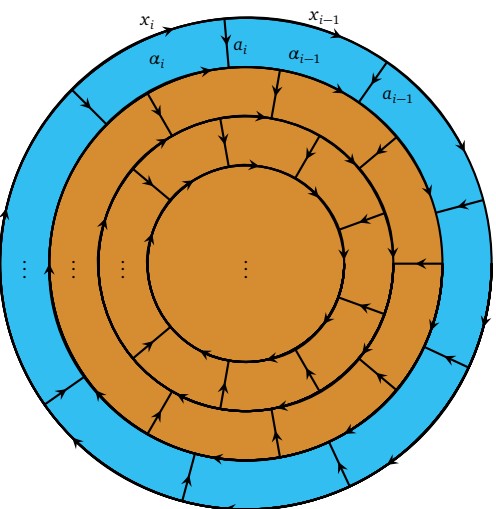

Figure 8: Trivalent lattice of 2D symmetry-enriched string-net model. The blue region corresponds to the boundary of the model.

ambiguity, we assume that the empty region outside the disk is a big plaquette on which lives a "ghost" spin $\mathbf{g}_{\text{empty}}$. We set the "ghost" spin $\mathbf{g}_{\text{empty}} = 1$ as a choice of boundary conditions. This choice corresponds the convention that the empty region below the horizontal line in Fig. 1a is taken to be the identity domain. Under this convention, all $P_l$ can be defined in the same way. All terms in (65) commute.

We would like to find the ground-state subspace of $H_{\text{disk}}$. We will see that it is highly degenerate, and the degeneracy comes from the states of boundary spins. First of all, we note that, in the ground-state subspace, the requirements $A_v = P_l = 1$ on the boundary spins (in the blue region) are exactly those we impose when building up the Hilbert space of our 1D model (Sec. 2.2). For the convenience of later discussions, we define a subspace $\mathcal{H}_{A_v = P_l = 1}$ in which $A_v = P_l = 1$ are fulfilled for all $v$'s and $l$'s. The ground-state space $\mathcal{H}_{\text{GS}} \subset \mathcal{H}_{A_v = P_l = 1}$. To find $\mathcal{H}_{\text{GS}}$, we note that all terms in $H_{\text{disk}}$ does not change the boundary spins $\{\alpha_i, a_i, x_i\}$. Then, we can diagonalize $H_{\text{disk}}$ in the subspace with fixed $\{\alpha_i, a_i, x_i\}$. We claim that, for a fixed set $\{\alpha_i, a_i, x_i\}$ that satisfies the requirements $A_v = P_l = 1$, the ground-state subspace is one-dimensional. That is, the ground-state subspace

$$\mathcal{H}_{\text{GS}} = \bigoplus_{\{\alpha_i, a_i, x_i\}} \mathcal{H}^{\text{GS}}_{\{\alpha_i, a_i, x_i\}}, \tag{66}$$

where each space $\mathcal{H}^{\text{GS}}_{\{\alpha_i, a_i, x_i\}}$ is one-dimensional.

We need to show $\mathcal{H}^{\text{GS}}_{\{\alpha_i, a_i, x_i\}}$ is one-dimensional for given $\{\alpha_i, a_i, x_i\}$ that statisfy $A_v = P_l = 1$. To simplify the calculation, we make use of the fact that the SESN bulk ground state is a fixed-point wave function, such that topological quantities, specifically ground-state degeneracy for our purpose, are invariant if we add or remove vertices, links or plaquettes in the bulk (orange region in Fig. 8). For detailed discussions about this property, readers may consult Ref. [57] (strictly speaking, only the original string-net model was discussed there, but we believe it can be straightforwardly generalized to the SESN model). With this property, we choose a simple graph, shown in Fig. 9(a), which contains only one bulk plaquette. On this lattice, besides the boundary spins $\{\alpha_i, a_i, x_i\}$, the only bulk degrees of freedom are the link spins $\{y_i\}$ and a central plaquette spin $\mathbf{g}_p$. A general basis state is labeled as $|\{\alpha_i, a_i, x_i, y_i, \mathbf{g}_p\}\rangle$. In following discussion, we will restrict ourselves in the subspace $\mathcal{H}_{A_v = P_l = 1}$. In this subspace, the Hamiltonian $H_{\text{disk}}$ effectively contains only one $B_p$ term associated with the central plaquette.

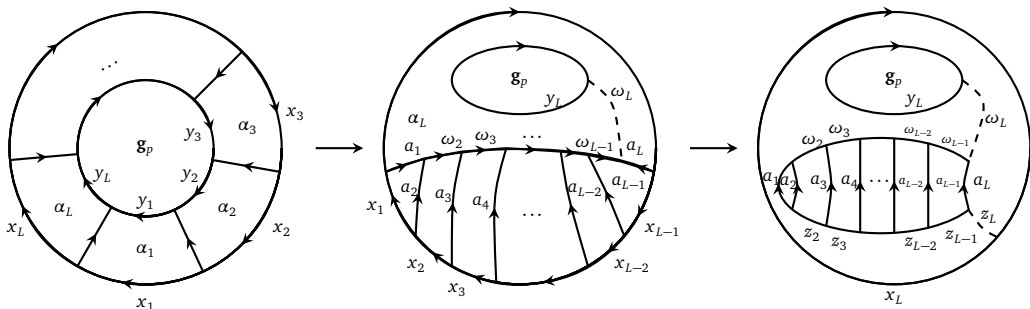

Figure 9: (a) Lattice with only one bulk plaquette. (b) and (c) States in $\mathcal{H}_{A_v=P_l=1}$ after proper $F$ moves. The dashed lines $w_L$ and $z_L$ correspond to the trivial string.

To proceed, we perform "basis transformations" for $\mathcal{H}_{A_v=P_l=1}$. More precisely, we will perform a transformation within the space

$$\mathcal{H}_{\{\alpha_i,\mathbf{g}_p\}} = \text{span}\{|\{\alpha_i, a_i, x_i, y_i, \mathbf{g}_p\}\rangle|a_i, x_i, y_i \in \mathcal{C}_G, A_v = P_l = 1, \forall\, v, l\}, \tag{67}$$

where $\{\alpha_i\}$ and $\mathbf{g}_p$ are fixed, and

$$\mathcal{H}_{A_v=P_l=1} = \bigoplus_{\{\alpha_i,\mathbf{g}_p\}} \mathcal{H}_{\{\alpha_i,\mathbf{g}_p\}}. \tag{68}$$

Because of the constraint $A_v = 1$ for all $v$'s, the states in $\mathcal{H}_{\{\alpha_i,\mathbf{g}_p\}}$ can be viewed of as fusion states of objects $\{a_i, x_i, y_i\}$. In this view, we can then perform $F$ moves which transform $\mathcal{H}_{\{\alpha_i,\mathbf{g}_p\}}$ into a different basis. Such transformation is not a standard basis transformation on lattice, as the underlying lattice structure is modified. However, it works well for our purpose of counting dimensions of the constrained Hilbert space $\mathcal{H}_{\{\alpha_i,\mathbf{g}_p\}}$. First, we perform $F$ moves and turn Fig. 9(a) into Fig. 9(b). Basis vectors in Fig. 9(b) are denoted as $|\alpha_i, a_i, x_i, w_i, y_L, \mathbf{g}_p\rangle$, subject to $A_v = P_l = 1$. An important feature is that the total fusion channel $w_L$ of $\{a_i\}$ (dashed line in Fig. 9(b)) must be 1. To see that, we recall a basic diagrammatic relation in fusion category theory [42]:

$$a \left\langle\!\!\!\begin{array}{c} c \\[6pt] \\[6pt] c' \end{array}\!\!\!\right\rangle b \;=\; \delta_{c,c'}\Big\uparrow. \tag{69}$$

The perimeter of the central plaquette is a special case of this relation with $c' = 1$ and $c = w_L$. Hence, $w_L = 1$. Then, the $y_L$ string decouples from the rest strings.

Now we make two claims for states in Fig. 9(b): (i) $\{w_i\}$ are completely fixed by $\{\alpha_i, a_i, x_i\}$ due to constraints $A_v = P_l = 1$ and thereby are redundant and (ii) the remaining degeneracy due to $\mathbf{g}_p$ and $y_L$ is completely lifted by the $B_p$ term associated with the central plaquette in $H_{\text{disk}}$. Under these two claims, we then immediately have $\mathcal{H}_{\{\alpha_i,a_i,x_i\}}^{\text{GS}}$ is one-dimensional.

The first claim can be shown by performing additional $F$ moves into Fig. 9(c). Note that these $F$ moves do not touch on $\{w_i\}$. Accordingly, if $\{w_i\}$ are fully fixed by other spins in Fig. 9(c), so are they in Fig. 9(b). Indeed, in the basis of Fig. 9(c), we have $w_i = z_i$ for every $i$. This is obtained by repeatedly applying the relation (69) to Fig. 9(c).

Given the first claim, we then have all valid states in $\mathcal{H}_{A_v=P_l=1}$ with fixed $\{\alpha_i, a_i, x_i\}$ form the following space

$$\mathcal{H}_{\{\alpha_i,a_i,x_i\}} = \text{span}\{|y_L, \mathbf{g}_p\rangle|y_L \in \mathcal{C}_G,\ \mathbf{g}_p = \alpha_L \mathbf{g}_{y_L}\}, \tag{70}$$

where the condition $\mathbf{g}_p = \alpha_L \mathbf{g}_{y_L}$ follows from the constraint $P_l = 1$. We note that $\mathcal{H}_{\{\alpha_i,a_i,x_i\}}$ is always $|\mathcal{C}_G|$-dimensional. The action of $H_{\text{disk}} = -B_p$ is closed in $\mathcal{H}_{\{\alpha_i,a_i,x_i\}}$. To prove the second claim, we need to calculate the ground state degeneracy inside $\mathcal{H}_{\{\alpha_i,a_i,x_i\}}$. We recall that $B_p$ is a projector, i.e., $B_p^2 = B_p$. Hence, the ground states have $B_p$ eigenvalue 1 and the excited states have $B_p$ eigenvalue 0. Then, the ground state degeneracy is given by $\text{Tr}(B_p)$. We show in Appendix B that $\text{Tr}(B_p) = 1$ in $\mathcal{H}_{\{\alpha_i,a_i,x_i\}}$ for arbitrary $\{\alpha_i, a_i, x_i\}$, i.e., $\mathcal{H}^{\text{GS}}_{\{\alpha_i,a_i,x_i\}}$ is one-dimensional.

To summarize, we have shown that the ground-state space $\mathcal{H}_{\text{GS}}$ of $H_{\text{disk}}$ in (65) is of the form (66), with $\mathcal{H}^{\text{GS}}_{\{\alpha_i,a_i,x_i\}}$ being one-dimensional. That is, $\mathcal{H}_{\text{GS}}$ is fully described by the boundary spins $\{\alpha_i, a_i, x_i\}$ subject to the constraints $A_v = P_l = 1$ for all relevant vertices and links. To exactly match our 1D model, we introduce additional interaction between the boundary spins

$$H' = H_{1D} - \Delta \sum_{l_a} K_{l_a}, \tag{71}$$

where $H_{1D}$ is the 1D Hamiltonian in Sec. 2.4, and $\Delta$ is a large positive number. The sum in the second piece runs over all links $l_a$ that $\{a_i\}$ lives. When acting on basis states, the operator $K_{l_a} = \delta(a_i, \bar{a}_{\alpha_{i-1}^{-1}\alpha_i})$, where $\bar{a}_{\mathbf{g}}$ is the selected object from $\mathcal{C}_{\mathbf{g}}$ discussed in Sec. 2.2 (we have added a bar in the notation to distinguish it from $a_i$ on links). In the limit $\Delta \to \infty$, this boundary theory matches exactly to our 1D model.

In the above discussions, we have focused on the Hilbert space and Hamiltonian, and have not touched on symmetry. The SESN model has an onsite $G$ group symmetry, while the 1D model is *not* symmetric under onsite $G$, instead is symmetric under $\mathcal{C}_G$. To understand this, let us apply $U^{\mathbf{g}}$ of (63) onto $\mathcal{H}_{\text{GS}}$. Let $|\{\alpha_i, a_i, x_i\}\rangle$ be the state in $\mathcal{H}^{\text{GS}}_{\{\alpha_i,a_i,x_i\}}$. Due to the constraints $P_l = 1$ and the boundary condition $\mathbf{g}_{\text{empty}} = 1$, we have $x_i \in \mathcal{C}_{\alpha_i}$ and $a_i \in \mathcal{C}_{\alpha_{i-1}^{-1}\alpha_i}$. Then, $U^{\mathbf{g}}|\{\alpha_i, a_i, x_i\}\rangle \sim |\mathbf{g}\alpha_i, a_i, x_i'\rangle$ with $x_i' \in \mathcal{C}_{\mathbf{g}\alpha_i}$. On the one hand, since $x_i' \notin \mathcal{C}_{\alpha_i}$, the ground-state $|\{\alpha_i, a_i, x_i\}\rangle$ transforms nontrivially under $G$, making it broken in some sense. On the other hand, the choice of $\{x_i'\}$ is not unique. To fix this ambiguity, we think of $U^{\mathbf{g}} = \prod_p U_p^{\mathbf{g}}$ as a union of all plaquettes and take a string $s \in \mathcal{C}_{\mathbf{g}}$ as its termination on the boundary. This termination means that, after applying $U^{\mathbf{g}}$, we further fuse $s$ onto $\{x_i\}$ from outside. Let us denote the string fusion operator as $B_0^s$, such that the combination sends $|\{\alpha_i, a_i, x_i\}\rangle$ to the state $B_0^s U^{\mathbf{g}}|\{\alpha_i, a_i, x_i\}\rangle$. One may notice that it is similar to the $B_p$ operator in the Hamiltonian, except that $U^{\mathbf{g}}$ has a left group action and $B_0^s$ fuses the $s$ string from outside of the plaquette in comparison to "right action" and "fusion from inside" for $B_p$ in the Hamiltonian. The collection $\{B_0^s U^{\mathbf{g}_s}\}$ with $s$ running over all simple objects in $\mathcal{C}_G$ are exactly the category symmetries discussed in Sec. 2.3.

Finally, we remark that while we have taken the limit $\Delta \to \infty$ in (71), one may also set $\Delta = 0$ and allow $\{a_i\}$ to fluctuate more freely, such that different boundary theories result. In addition, we only consider the case that bulk is in the ground state. If the bulk contains a topological defect, including both anyon excitations and $G$ symmetry defects, there must be a corresponding anti-defect on the boundary. (Note that it is enough to consider only one topological defect in the bulk. Multiple defects can always be fused into one.) This will make at least one of the constraints $A_v = P_l = 1$ to be violated at the boundary, corresponding to insertion of twisted boundary conditions associated with the category symmetry $\mathcal{C}_G$ in the 1D systems.

# 5 Summary and outlook

In summary, we have constructed a 1D quantum lattice model that explicitly displays category symmetry $\mathcal{C}_G$. The model can be viewed as an interpolation between the anyon chain model and edge model of 2D bosonic SPTs, and as an edge model of 2D bosonic SETs. Our numerical results show that the category symmetry constrains the model to the extent that it has a large likelihood to be quantum critical. Hence, this model, with different input categories and tuning parameters, is a good source for studying gapless phases. It is clear that more numerical effort is desired.

We discus a few possible future directions.

1. One may generalize our model to $G$-graded super or spin unitary fusion category (we notice that a related work is done in Ref. [58]). Super fusion category describes defects in fermionic systems, and spin fusion category is the corresponding category after gauging fermion parity. [59,60] Our model can be readily generalized to spin fusion category, which has no difference to the usual unitary fusion category except that it has a special simple object, the fermion $\psi$. To make a connection to fermionic SPT/SET edges, one needs to find a way to ungauge the fermion parity, or equivalently gauge the dual symmetry $U(\psi)$. This gauging procedure has been worked out in Ref. [38] in the example of Ising fusion category (the simplest spin fusion category). It is interesting to work out the general case and understand the connection to fermionic SPT/SET edges. (*Note added*. During the publication process of our work, we noticed a few recent works [61, 62] on 1+1 systems with fermionic categorical symmetries.)

2. Another generalization is to make the variable $x_i$ valued in a module category $\mathcal{M}$ over a fusion category $\mathcal{C}$ [16]. It is known that a general way to terminate the string-net model at the boundary is to use module category [63]. The recent study on duality of category symmetry in Ref. [34] precisely uses this language. The essence of having a $G$-grading structure in the input data $\mathcal{C}_G$ of our model is to enable a partial ungauging of the category symmetry. We expect that generalization to module category may help to ungauge general category symmetry in our model, which is essentially the duality discussed in Ref. [34].

3. $\mathcal{C}_G$ serves both as the input data and as the category that characterizes the symmetries of our model. In principle, one may make use of the Drinfeld center $\mathcal{Z}(\mathcal{C}_G)$, which describes all the anyons in the SET bulk after gauging $G$. For example, in the case that $\mathcal{C}_G = \mathcal{C}_{\text{Ising}}$ as input, the gauged SET bulk is characterized by $\mathcal{C}_{\text{Ising}} \times \overline{\mathcal{C}_{\text{Ising}}}$. In our construction, we only make use of $\mathcal{C}_{\text{Ising}}$ to constrain the low-energy physics, while $\overline{\mathcal{C}_{\text{Ising}}}$ has not be explicitly used. It is interesting to study how to construct models with the larger category symmetry $\mathcal{Z}(\mathcal{C}_G)$ manifested.

4. It is also interesting to extend this construction to higher dimensions. In this perspective, one needs to make use of higher fusion categories [20–22].

# Acknowledgments

We are grateful to Liang Kong, Tian Lan, Jiucai Wang, Qing-Rui Wang, Wenjie Xi, Shizhong Zhang and Cuo-Yi Zhu for enlightening discussions.

**Funding information** This work was supported by Research Grants Council of Hong Kong (GRF 11300819 and GRF 17311322) and National Natural Science Foundation of China (Grant No. 12222416). S.N. is also supported in part by Research Grants Council of Hong Kong under GRF 14302021 and NSFC/RGC Joint Research Scheme No. N-CUHK427/18, and by Direct Grant No. 4053416 from the Chinese University of Hong Kong.

## A Symmetry and Hamiltonian

In this appendix, we give a derivation of the explicit expression (6) of the symmetry $U(y_{\mathbf{h}})$. We also explicitly show that the Hamiltonian (8) is invariant under $U(y_{\mathbf{h}})$.

### A.1 Derivation of Eq. (6)

The graphical representation of $U(y_{\mathbf{h}})$ is shown in Fig. 2. Under the action of $U(y_{\mathbf{h}})$, the domain variables $\alpha_i$ are simultaneously mapped to $\mathbf{h}\alpha_i$. Since $\alpha_i^{-1}\alpha_{i+1}$ is unchanged, the domain wall defect $a_i$ keeps invariant under $U(y_{\mathbf{h}})$. Meanwhile, the variables $x_i$ will be mapped to other variables $x_i'$

$$U(y_{\mathbf{h}}) : |\{\alpha_i, x_i\}\rangle \rightarrow |\{\mathbf{h}\alpha_i, x_i'\}\rangle. \tag{A.1}$$

In general, $U(y_{\mathbf{h}})|\{\alpha_i, x_i\}\rangle$ is a linear superposition of $|\{\mathbf{h}\alpha_i, x_i'\}\rangle$. Below we show that the matrix element $\langle\{\mathbf{h}\alpha_i, x_i'\}|U(y_{\mathbf{h}})|\{\alpha_i, x_i\}\rangle$ is given by (6). The derivation is divided into four steps, as follows. Note that this derivation is equivalent to that for the usual anyon-chain models [29].

1. Add a trivial line connecting $y_{\mathbf{h}}$ and $x_{i+1}$ as in (A.2) and perform an $F$ move which would give an amplitude $\left[(F_{y_{\mathbf{h}}}^{y_{\mathbf{h}} x_{i+1} \overline{x_{i+1}}})^\dagger\right]_1^{x_{i+1}'} = \sqrt{\dfrac{d_{x_{i+1}'}}{d_{y_{\mathbf{h}}} d_{x_{i+1}}}} \delta_{y_{\mathbf{h}} x_{i+1} x_{i+1}'}$. Here, $\delta_{y_{\mathbf{h}} x_{i+1} x_{i+1}'} = N_{y_{\mathbf{h}} x_{i+1}}^{x_{i+1}'} = 0$ or 1. Summation over $x_{i+1}'$ is not shown.

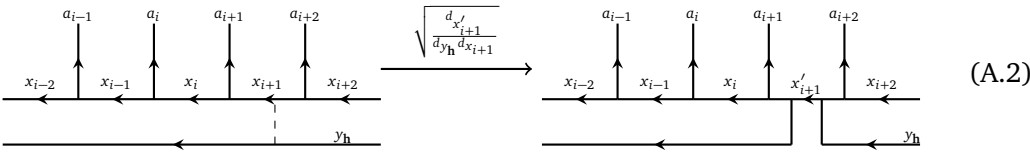

$$\tag{A.2}$$

2. Perform a $F$ move associated with the three defects $y_{\mathbf{h}}, x_i, a_{i+1}$, with $x_{i+1}'$ viewed as the total fusion channel, as in (A.3). We call this procedure "sliding $y_{\mathbf{h}}$ across $a_{i+1}$". It gives an amplitude $\left[(F_{x_{i+1}'}^{y_{\mathbf{h}}, x_i, a_{i+1}})^\dagger\right]_{x_{i+1}}^{x_i'}$.

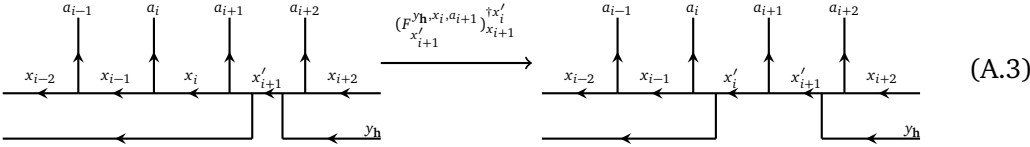

$$\tag{A.3}$$

3. Continue the second step, and keep sliding $y_\mathbf{h}$ across the rest $a_j$, as in (A.4). This gives the amplitude $\left[\prod_{j\neq i,i+1}(F^{y_\mathbf{h},x_j,a_{j+1}}_{x'_{j+1}})^{\dagger x'_j}_{x_{j+1}}\right](F^{y_\mathbf{h},x_{i+1},a_{i+2}}_{x''_{i+2}})^{\dagger x'_{i+1}}_{x_{i+2}}$.

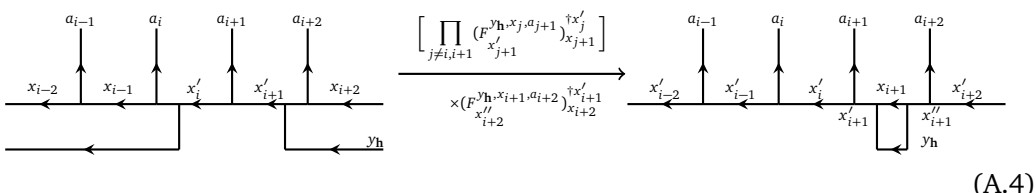

(A.4)

4. Shrink the "bubble" as in (A.5) which gives a coefficient $\sqrt{\dfrac{d_{x_{i+1}}d_{y_\mathbf{h}}}{d_{x'_{i+1}}}}$ and imposes the condition $x'_{i+1} = x''_{i+1}$.

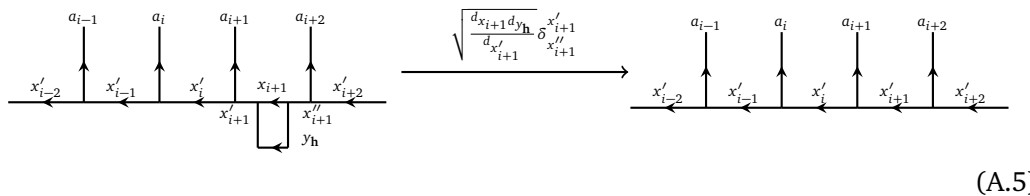

(A.5)

Combining all the steps and multiplying all the amplitudes, we obtain Eq. (6).

## A.2  Hamiltonian is symmetric under $U(y_\mathbf{h})$

Now we show that the Hamiltonian (8) is symmetric under $U(y_\mathbf{h})$ (6). Specifically, we show $H_i U(y_\mathbf{h}) = U(y_\mathbf{h})H_i$ when acting on any state. The graphical representation of $U(y_\mathbf{h})$ in Fig. 2 has the advantage of being basis independent. We will make use of this and mainly work in the basis (9). We will act $H_i$ and $U(y_\mathbf{h})$ on an arbitrary state in different orders, and compare the final sates, which turn to be the same.

On the one hand,

$$U(y_\mathbf{h})H_i \left|\begin{matrix}a_i & a_{i+1}\\ \alpha_{i-1} & \alpha_i & \alpha_{i+1}\\ x_{i-1} & x_i & x_{i+1}\end{matrix}\right\rangle = \sum_{z_i}\left[F^{x_{i-1}a_ia_{i+1}}_{x_{i+1}}\right]^{z_i}_{x_i} U(y_\mathbf{h})H_i \left|\begin{matrix}a_i & a_{i+1}\\ \alpha_{i-1} & \alpha_i & \alpha_{i+1}\\ z_i \\ x_{i-1} & x_{i+1}\end{matrix}\right\rangle \qquad (A.6)$$

$$= \sum_{\alpha'_i}\sum_{z_i} w^{z_i}_{\alpha_i^{-1}\alpha'_i}\left[F^{x_{i-1}a_ia_{i+1}}_{x_{i+1}}\right]^{z_i}_{x_i} U(y_\mathbf{h}) \left|\begin{matrix}a'_i & a'_{i+1}\\ \alpha_{i-1} & \alpha'_i & \alpha_{i+1}\\ z_i \\ x_{i-1} & x_{i+1}\end{matrix}\right\rangle$$

$$= \sum_{\alpha'_i}\sum_{z_i}\sum_{\{x'_j|j\neq i\}} w^{z_i}_{\alpha_i^{-1}\alpha'_i}\left(F^{x_{i-1}a_ia_{i+1}}_{x_{i+1}}\right)^{z_i}_{x_i} U^{\{x'_i\}}_{\{x_i\},y_\mathbf{h},z_i} \left|\begin{matrix}a'_i & a'_{i+1}\\ \mathbf{h}\alpha'_i \\ \mathbf{h}\alpha_{i-1} & z_i & \mathbf{h}\alpha_{i+1}\\ x'_{i-1} & x'_{i+1}\end{matrix}\right\rangle,$$

where we have used the basis transformation (9) in the first line, and the definition (10) of $H_i$ in th second line. The coefficient in the last line is

$$U^{\{x'_i\}}_{\{x_i\},y_\mathbf{h},z_i} = \left(F^{y_\mathbf{h},x_{i-1},z_i}_{x'_{i+1}}\right)^{\dagger x'_{i-1}}_{x_{i+1}} \prod_{j\neq i,i+1}\left(F^{y_\mathbf{h},x_j,a_{j+1}}_{x'_{j+1}}\right)^{\dagger x'_j}_{x_{j+1}}, \qquad (A.7)$$

which is obtained in the same way as Appendix A.1. On the other hand,

$$
H_i U(y_{\mathbf{h}}) \left| \begin{matrix} a_i & a_{i+1} \\ \alpha_{i-1} \;\; \alpha_i \;\; \alpha_{i+1} \\ x_{i-1} \;\; x_i \;\; x_{i+1} \end{matrix} \right\rangle = \sum_{z_i} \left[ F^{x_{i-1} a_i a_{i+1}}_{x_{i+1}} \right]^{z_i}_{x_i} H_i U(y_{\mathbf{h}}) \left| \begin{matrix} a_i \;\; \alpha_i \;\; a_{i+1} \\ \alpha_{i-1} \;\; z_i \;\; \alpha_{i+1} \\ x_{i-1} \;\; x_{i+1} \end{matrix} \right\rangle
\tag{A.8}
$$

$$
= \sum_{z_i} \sum_{\{x'_j | j \neq i\}} \left[ F^{x_{i-1} a_i a_{i+1}}_{x_{i+1}} \right]^{z_i}_{x_i} U^{\{x'_i\}}_{\{x_i\}, y_{\mathbf{h}}, z_i} H_i \left| \begin{matrix} a_i \;\; \mathbf{h}\alpha_i \;\; a_{i+1} \\ \mathbf{h}\alpha_{i-1} \;\; z_i \;\; \mathbf{h}\alpha_{i+1} \\ x'_{i-1} \;\; x'_{i+1} \end{matrix} \right\rangle
$$

$$
= \sum_{\alpha'_i} \sum_{z_i} \sum_{\{x'_j | j \neq i\}} \left( F^{x_{i-1} a_i a_{i+1}}_{x_{i+1}} \right)^{z_i}_{x_i} U^{\{x'_i\}}_{\{x_i\}, y_{\mathbf{h}}, z_i} w^{z_i}_{\alpha_i^{-1} \alpha'_i} \left| \begin{matrix} \mathbf{h}\alpha'_i \\ \mathbf{h}\alpha_{i-1} \;\; z_i \;\; \mathbf{h}\alpha_{i+1} \\ x'_{i-1} \;\; x'_{i+1} \end{matrix} \right\rangle,
$$

where we have used $(\mathbf{h}\alpha_i)^{-1}(\mathbf{h}\alpha_{i+1}) = \alpha_i^{-1}\alpha_{i+1}$. Comparing (A.6) and (A.8), we see the final expressions are exactly the same. As the initial state and $i$ are arbitrary, we have proven $HU(y_{\mathbf{h}}) = U(y_{\mathbf{h}})H$ for any $y_{\mathbf{h}}$.

# B  Proof of $\mathrm{Tr}(B_p) = 1$ in $\mathcal{H}_{\{\alpha_i, a_i, x_i\}}$

In this appendix, we show that $\mathrm{Tr}(B_p) = 1$ in the space $\mathcal{H}_{\{\alpha_i, a_i, x_i\}}$ with given $\{\alpha_i, a_i, x_i\}$. We will represent a state $|\Psi\rangle$ in $\mathcal{H}_{\{\alpha_i, a_i, x_i\}}$ graphically as

$$
|\Psi\rangle = \left| \begin{matrix} y_L \\ \mathbf{g}_p \end{matrix} \; \alpha_L \right\rangle ,
\tag{B.1}
$$

where $y_L$ can be any simple object in $\mathcal{C}_G$, $\mathbf{g}_p = \alpha_L \mathbf{g}_{y_L}$, and other spins on the lattice (Fig. 9(b)) are omitted as $B_p$ does not act on them. Since $\mathbf{g}_p$ is fixed by $y_L$ and $\alpha_L$, the dimension of $\mathcal{H}_{\{\alpha_i, a_i, x_i\}}$ is $|\mathcal{C}_G|$. The term $B_p$ is defined as $B_p = \frac{1}{D^2} \sum_{s \in \mathcal{C}_G} d_s B_p^s \tilde{U}_p^{\mathbf{g}_s}$, where $D = \sqrt{\sum_s d_s^2}$ and

$$
B_p^s \tilde{U}_p^{\mathbf{g}_s} \left| \begin{matrix} y_L \\ \mathbf{g}_p \end{matrix} \; \alpha_L \right\rangle = B_p^s \left| \begin{matrix} y_L \\ \mathbf{g}_p \mathbf{g}_s \end{matrix} \; \alpha_L \right\rangle = \left| \begin{matrix} y_L \\ s \\ \mathbf{g}_p \mathbf{g}_s \end{matrix} \; \alpha_L \right\rangle = \sum_{y'_L} N^{y'_L}_{y_L, s} \left| \begin{matrix} y'_L \\ \mathbf{g}_p \mathbf{g}_s \end{matrix} \; \alpha_L \right\rangle .
\tag{B.2}
$$

In the last equation, we have fused $y_L$ and $s$ strings, with $N^{y'_L}_{y_L, s} = 0, 1$ being the fusion coefficient. Note that individual action of $\tilde{U}_p^{\mathbf{g}_s}$ or $B_p^s$ goes out of the space $\mathcal{H}_{\{\alpha_i, a_i, x_i\}}$. We have omitted arrows of the strings for simplicity, which can be easily restored.

We calculate $\mathrm{Tr}(B_p)$ as follows:

$$
\begin{aligned}
\mathrm{Tr}(B_p) &= \sum_{y_L \in \mathcal{C}_G} \left\langle \alpha_L \left( \overset{y_L}{\mathbf{g}_p} \right) \middle| B_p \middle| \left( \overset{y_L}{\mathbf{g}_p} \right) \alpha_L \right\rangle \\
&= \sum_{y_L \in \mathcal{C}_G} \sum_{s \in \mathcal{C}_G} \frac{d_s}{D^2} \left\langle \alpha_L \left( \overset{y_L}{\mathbf{g}_p} \right) \middle| B_p^s \tilde{U}_p^{\mathbf{g}_s} \middle| \left( \overset{y_L}{\mathbf{g}_p} \right) \alpha_L \right\rangle \\
&= \sum_{y_L \in \mathcal{C}_G} \sum_{s \in \mathcal{C}_G} \sum_{y_L' \in \mathcal{C}_G} \frac{d_s}{D^2} N_{y_L,s}^{y_L'} \left\langle \alpha_L \left( \overset{y_L}{\mathbf{g}_p} \right) \middle| \left( \overset{y_L'}{\mathbf{g}_p \mathbf{g}_s} \right) \alpha_L \right\rangle \\
&= \sum_{y_L \in \mathcal{C}_G} \sum_{s \in \mathcal{C}_G} \sum_{y_L' \in \mathcal{C}_G} \frac{d_s}{D^2} N_{y_L,s}^{y_L'} \delta_{y_L,y_L'} \delta_{\mathbf{g}_s,1} \\
&= \sum_{y_L \in \mathcal{C}_G} \sum_{s \in \mathcal{C}_0} \frac{d_s}{D^2} N_{y_L,s}^{y_L} = \sum_{y_L \in \mathcal{C}_G} \frac{d_{y_L}^2}{D^2} = 1 \,.
\end{aligned}
\tag{B.3}
$$

In the third line, we have inserted Eq. (B.2). In the last line, we have used $d_a = d_{\bar{a}}$, $N_{ab}^c = N_{a\bar{c}}^{\bar{b}}$ and $d_a d_b = \sum_c d_c N_{ab}^c$ for any $a, b, c \in \mathcal{C}_G$, such that $\sum_s d_s N_{y_L,s}^{y_L} = \sum_s d_{\bar{s}} N_{y_L, \overline{y_L}}^{\bar{s}} = d_{y_L}^2$. Note that if $N_{y_L,s}^{y_L} \neq 0$, we must have $s \in \mathcal{C}_0$ due to the $G$-grading structure in $\mathcal{C}_G$.

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
