# Peer review of "Building 1D lattice models with $G$-graded fusion category"

_SciPost Physics, doi:SciPost Phys. 17, 125 (2024)_

## Round 1 · Referee Report · Anonymous (Referee 1) · 2024-2-22

Report

This paper gives a systematic construction of 1D lattice models using graded fusion category degrees of freedom. In particular, the authors take the graded fusion category data, define a constrained Hilbert space out of it, and determine the Hamiltonian that satisfies the graded fusion category symmetry. Several example Hamiltonians are then studied numerically and shown to contain gapless regions in the phase diagram. This is an interesting work, especially given the recent interest in categorical symmetries. Using the protocol given in this paper, one can systematically construct models with categorical symmetry that reflect the edge physics of 2D symmetry enriched phases. The paper is very carefully written and is a nice addition to the literature. I only have one minor comment: This paper https://arxiv.org/abs/2110.12882 seems to be on a related topic, although the result seems to be a subset of that in this paper. Can the authors comment on their relation?

  • validity: high
  • significance: high
  • originality: high
  • clarity: top
  • formatting: perfect
  • grammar: excellent

Author:  Chenjie Wang  on 2024-09-06  [id 4742]

(in reply to Report 1 on 2024-02-22)

We thank the referee for a high evaluation of our work. Regarding the relation to arXiv:2110.12882, we mention two differences. First, it constructs 1+1D lattice models with a gapped, symmetric and unique ground state, i.e., SPT phases with fusion category symmetry. The models of arXiv:2110.12882 are exactly solvable Hamiltonians of local commutative projectors. On the other hand, the Hamiltonians in our work do not consist of commuting projectors and in general cannot be solved analytically. Also, the ground states may be gapless or break fusion category symmetry. Second, to admit a gapped symmetric unique ground state (i.e., be anomaly-free), the fusion category is highly restricted: it must be the representation category of a Hopf algebra. Accordingly, arXiv:2110.12882 uses the language of Hopf algebra for the model construction. On the other hand, we do not require the fusion category to be anomaly-free, so our construction works for general $G$-graded fusion category. We have briefly described these differences in the revised manuscript (see the sentences around line 111).

Anonymous on 2024-09-21  [id 4795]

(in reply to Chenjie Wang on 2024-09-06 [id 4742])

Thanks for the reply. I think the paper is ready for publication now.

---

## Round 1 · Referee Report · Anonymous (Referee 2) · 2024-6-24

Strengths

1) Fills in an obvious hole in the literature on non-invertible symmetries, lattice models, and boundaries of topological order.
2) Clearly written with some simple, analytic examples.
3) Presents both analytic and numerical results.

Weaknesses

There are some points that I think can be clarified. See comments/questions below.

Report

This paper presents a construction for an interesting class of models describing boundary theories for symmetry-enriched topological orders. These models are amenable to numerical study, and provide a simple generalization of anyon chain models, which have been useful for studying properties of gapless theories with non-invertible symmetries. The quality of the paper is high, but I would like to receive answers to the following comments/questions along with appropriate edits to the paper before recommending it for publication:

1) Very small comment on p 1: "string operators associated with moving abelian anyons in the 2D toric code model" may not be the best way to describe 1-form symmetries, because string operators that move anyons are open strings, which do not commute with the Hamiltonian and do not leave the ground state invariant. Perhaps it is better to use "closed string operators in the 2D toric-code model."

2) The paper draws a connection between the boundaries of SETs and their 1D lattice models. Can the connection be made more concrete? For example, a 2+1D SET is described by a set of data including the anyon permutation by G and an H^2(G,A) class. These pieces of data are important for boundary physics because they determine whether or not the boundary can be G-symmetrically gapped. How are these pieces of data read off from the description of the symmetry in 1+1d?

3) Can you comment more on the choice of $a_i$ and the qualitative effects on the physics of the lattice model? In the Ising example, what would happen if you chose $a_0=\psi$ instead of $a_0=1$?

4) Can you comment on the different choices of $\{w_h^z\}$? It seems that if you set $w_0^z=1,w_{h\neq 0}^z=0$ for all sites then you always a $G$ SSB state? Maybe you can add some comments on why the $H^3(G,U(1))$ class only affects $w_g^0$ in the $\mathbb{Z}_2$ example (does a similar result hold for more general $G$?)

5) Right above eq 38, you mention "We are interested in the cases that the models are gapless, which can be described by conformal field theory (CFT)." How do you know that the gapless regions of the phase diagram can always be described by a CFT? In some points in Fig 3, you certainly don't get CFT behavior because you have quadratic dispersion.

6) Right above Sec 3.7, do you expect that for at least some choices of parameters i.e. for some $\{w_h^z\}$ you get decoupled Fib and $\mathbb{Z}_2$ theories? i.e. something like a decoupled golden chain and a trivial $\mathbb{Z}_2$ paramagnet.

7) To be clear, the F moves below Eq 65 are different from the F symbols of the (graded) fusion category? Because they are F symbols of the G-crossed BTC.

8) Should I think of claim (ii) on line 685 as coming from the fact that even if we get an SSB Hamiltonian, the non-tensor-product structure of the Hilbert space projects out all of the ground states except for one? However, if we reduce to the SPT boundary i.e. we set $\mathcal{C}_0=\{1\}$, then we should get degeneracy because we can get SSB boundary theories and the boundary has a tensory product Hilbert space?

9) Regarding fermionic theories (outlook point 1), https://arxiv.org/pdf/2404.19004 might be of interest. It might also be worth citing https://arxiv.org/abs/2304.01262 and https://link.springer.com/article/10.1007/JHEP10(2023)053 in line 68

10) Regarding outlook point 2, I'm confused about why you need the $x_i$ to come from other module categories. You already seem to get access to different phases just by tuning the $\{w_h^z\}$ variables? And the reason for changing the module category is to access other 1+1D phases with the same graded fusion category symmetry. Does this mean that for a given set of $\{w_h^z\}$ for a fixed set $\{x_i\}$ you only get access to a single gapped phase (and gapless regions)? And in order to access other gapped phases you need to change $\{x_i\}$? For example, for the simple $G=\mathbb{Z}_2$ case with $\mathcal{C}_0=\{1\}$, you get both the trivial symmetric phase ($w_h^z=1$ for all $h,z$) and the SSB phase ($w_0^z=1,w_{h\neq 0}^z=0$). Can you comment more on which phases you expect to access given $\{x_i\}$?

11) Regarding outlook point 3, I'm not sure what you mean by maximal category symmetry here? For example, you write "More generally, one may expect a larger category symmetry $\mathcal{Z}(\mathcal{C}_G)$ in the gapless state of the model." However, $\mathcal{Z}(\mathcal{C}_G)$ is braided while the symmetry of a 1+1D system contains only fusion data? It seems that the maximal category symmetry should be another fusion category, not the center of the fusion category which is a braided tensor category?

12) You don't have to do this, but it would be interesting to compare the anomalous vs non-anomalous phase diagram, to make the claim about "larger likelihood to be gapless" more concrete. Here you only have the phase diagram in Fig 3 for the model with the anomalous Z2 symmetry.

Recommendation

Ask for minor revision

  • validity: top
  • significance: good
  • originality: high
  • clarity: high
  • formatting: excellent
  • grammar: excellent

Author:  Chenjie Wang  on 2024-09-06  [id 4743]

(in reply to Report 2 on 2024-06-24)

We thank the referee for a high evaluation on our paper and for considering it "clearly written''. Below are our replies to the comments and questions.

1) Very small comment on p 1: "string operators associated with moving abelian anyons in the 2D toric code model" may not be the best way to describe 1-form symmetries, because string operators that move anyons are open strings, which do not commute with the Hamiltonian and do not leave the ground state invariant. Perhaps it is better to use "closed string operators in the 2D toric-code model.”

Reply: Thanks for pointing it out. We have rephrased the presentation accordingly in the revised manuscript.

2) The paper draws a connection between the boundaries of SETs and their 1D lattice models. Can the connection be made more concrete? For example, a 2+1D SET is described by a set of data including the anyon permutation by G and an $H^2(G,A)$ class. These pieces of data are important for boundary physics because they determine whether or not the boundary can be G-symmetrically gapped. How are these pieces of data read off from the description of the symmetry in 1+1d?

Reply: To address this question, we begin with a general description of 2+1D bosonic SET phases with a unitary symmetry $G$. (We consider SPT phases a special case of SET phases.) A general SET phase is described by the set of data $(\mathcal{D}, \rho, \omega_2,\nu_3)$, where $\mathcal{D}$ is a unitary modular tensor category, $\rho$ describes the anyon permutation, $\omega_2$ is a 2-cocycle in $H^2(G,A)$, and the 3-cocycle $\nu_3\in H^3(G,U(1))$ describes the stacking of 2D SPT phases. Whether the boundary of an SET can be gapped out while preserving the symmetry group $G$ depends all the data in $(\mathcal{D},\rho,\omega_2,\nu_3)$. First of all, to be gapable, it is required that there exists at least one Lagrangian algebra $\mathcal{A}$ in $\mathcal{D}$. Equivalently, $\mathcal{D}$ has to be the Drinfeld center of a unitary fusion category $\mathcal{C}_0$, $\mathcal{D}=Z(\mathcal{C}_0)$. To further require the gapped boundary be $G$-symmetric, complicated conditions need to be imposed between $\mathcal{A}$ and $(\rho,\omega_2,\nu_3)$.

Meanwhile, for topological orders with gappable boundaries, i.e., $\mathcal{D}=Z(\mathcal{C}_0)$, there exists another description of SET phases. It is the $G$ extensions of the unitary fusion category $\mathcal{C_0}$ (see arxiv:0909.3140). After an $G$ extension, one has a $G$-graded fusion category $\mathcal{C}_G$ (which is introduced in the manuscript). Different $G$ extensions of $\mathcal{C}_0$ correspond to different SETs of $\mathcal{D}$. The information of $(\rho,\omega_2,\nu_3)$ are secretly encoded in the $G$-graded category $\mathcal{C}_G$. Lattice model realizations of SET phases based on $\mathcal{C}_G$ can be found in arxiv:1606.08482 and arXiv:1606.07816.

That being said, we see that the appropriate question is how to extract the SET data $(\rho,\omega_2,\nu_3)$ from the graded fusion category $\mathcal{C}_G$. Whether or not we are studying the SET in the bulk or on the boundary is irrelevant for this question. In general, extracting $(\rho,\omega_2,\nu_3)$ from $\mathcal{C}_G$ is not known beyond a few examples. In our paper, we use the language of $G$-graded fusion category for the description of 1+1D boundary models. One may wonder how to 1+1D models using the SET data $(\rho,\omega_2,\nu_3)$. This is a problem for future study.

3) Can you comment more on the choice of $a_i$ and the qualitative effects on the physics of the lattice model? In the Ising example, what would happen if you chose $a_0=\psi$ instead of $a_0=1$?

Reply: Different choices of ${a_i}$ give rise to distinct Hilbert spaces, different realizations of symmetry, and consequently, different models. The symmetry operator in Eq. 6 and Hamiltonian in Eq. 11 have explicit dependence on ${a_i}$. Indeed, the effect of choices of ${a_i}$ is an interesting problem to study. However, we haven't done any study yet, including the case of Ising fusion category.

4) Can you comment on the different choices of {$w^z_h$}? It seems that if you set $w^z_0=1,w^z_{h\neq 0}=0$ for all sites then you always a G SSB state? Maybe you can add some comments on why the $H^3(G,U(1))$ class only affects $w^0_g$ in the $Z_2$ example (does a similar result hold for more general $G$?)

Reply: We answer the three questions in order.

(i) To the first question, we are only able to make a general comment that ${w^{z}_{h}} $ are coupling parameters of the model, and different values of ${w_h^z}$ may put the model in different phases.

(ii) To the second question, the setting $w^z_0=1,w^z_{h\neq 0}=0$ for all $z$'s means that, taking the model of Ising fusion category as an example, the Hamiltonian Eq. 25 is a constant. Perhaps, the referee is thinking of the case that $w_{h\neq 0}^z=0$ while $w_0^z$ are different for different $z$'s. In this case, indeed, no terms in the Hamiltonian can flip the domain degrees of freedom ${\alpha_i}$. The Hamiltonian seems deep in a "ferromagnetic" or "anti-ferromagnetic" phase (i.e., $\Delta =0$ and $r =\pm \infty$ in Eq. 26). They indeed correspond to the spontaneous breaking of $G$ and also the fusion category symmetry $\mathcal{C}$. However, after a careful investigation, one will find that the ground state degeneracy is infinite in the thermodynamic limit. This large degeneracy can be lifted by introducing a small $\Delta$ (i.e., small $w_{h\neq 0}^z$), with which the magnetic ordering is then well defined. We will study the physics of spontaneous breaking of categorical symmetry in future works.

(iii) To the third question, that only $w_g^0$ is affected by $H^3(G,U(1))$ holds only for $G=Z_2$. For a 3-cocycle in $H^3(Z_2,U(1))$, only the element $ \nu_3(g,g,g) = -1$ and all others are 1 (under certain gauge choice). This means that the nontrivial $\nu_3(g,g,g)$ enter the Hamiltonian matrix elements given in Eq. (13) only when $\alpha_{i-1} = \alpha_{i+1} = g$ and $\alpha_i \neq \alpha_i'$. That is, $\nu_3$ matters only for $w_g^0$. For a generic group $G$, this condition does not hold.

5) Right above eq 38, you mention "We are interested in the cases that the models are gapless, which can be described by conformal field theory (CFT)." How do you know that the gapless regions of the phase diagram can always be described by a CFT? In some points in Fig 3, you certainly don't get CFT behavior because you have quadratic dispersion.

Reply: We are sorry for the inaccurate presentation which causes misunderstanding. The sentence cited in the question is only intended to convey that the gapless phases (or points) characterized by CFTs are our main interests. It does not imply that all gapless states of our model are described by CFTs. We appreciate this observation and have rephrased the sentence in the revised manuscript --- see discussions around line 417.

6) Right above Sec 3.7, do you expect that for at least some choices of parameters i.e. for some ${w^z_h}$ you get decoupled Fib and Z2 theories? i.e. something like a decoupled golden chain and a trivial Z2 paramagnet.

Reply: It is possible that one can get decoupled Fibonacci and $Z_2$ theories for certain choices of parameters. This would require the choice $a_0=\tau$ and $a_1=1$; instead, if $a_0=1$ and $a_1=\tau$, it means that Fibonacci defects $\tau$ is decorated on $Z_2$ domain walls, so that the two degrees of freedom are coupled. However, we do not anticipate that the decoupled theories resemble a system of a decoupled golden chain alongside a trivial ${Z}_2$ paramagnet. This is because the $SU(2)_3$ theory is equivalent to Fibonacci anyon stacked with a nontrivial ${Z}_2$ SPT state. Accordingly, we expect the decoupled $Z_2$ theory shall resemble the edge of $Z_2$ SPT. Nevertheless, this is speculation and numerical analysis is needed for confirmation.

7) To be clear, the F moves below Eq 65 are different from the F symbols of the (graded) fusion category? Because they are F symbols of the G-crossed BTC.

Reply: No, the $F$ moves below Eq. (65) [Eq. (68) of the revised manuscript] are the same as the $F$ symbols of the graded fusion category. The input category of the symmetry-enriched string-net model is the same $G$-graded fusion category as the one that we use for the 1D lattice models. To clarify, the input category is different from the $G$-crossed braided tensor category that characterizes the output SET phase of the symmetry-enriched string-net model.

8) Should I think of claim (ii) on line 685 as coming from the fact that even if we get an SSB Hamiltonian, the non-tensor-product structure of the Hilbert space projects out all of the ground states except for one? However, if we reduce to the SPT boundary i.e. we set $C_0={1}$, then we should get degeneracy because we can get SSB boundary theories and the boundary has a tensor product Hilbert space?

Reply: Claim (ii) on line 685 of the previous version is a statement on the degeneracy for a fixed set of parameters ${\alpha_i, a_i, x_i}$. Specifically, it states that, for a fixed set ${\alpha_i,a_i,x_i}$, the ground state degeneracy is 1. By ``fixing ${\alpha_i, a_i, x_i}$'', one may think of applying an appropriate fictitious Zeeman-like external field to break all symmetries explicitly, which then gives rise to a unique ground state. To obtain the full degenerate Hilbert space, we sum over all $\alpha_i, a_i$ and $x_i$ (i.e., given by Eq. 66 in the revised manuscript). This full ground-state space corresponds precisely to the Hilbert space of the 1D model under consideration. With additional interaction on the boundary (ie., Eq 71 in the revised manuscript0), spontaneously symmetry breaking may occur, depending on the details of the boundary Hamiltonian. Accordingly, the claim (ii) does not contradict with the SSB phenomenon of the SPT boundary.

9) Regarding fermionic theories (outlook point 1), https://arxiv.org/pdf/2404.19004 might be of interest. It might also be worth citing https://arxiv.org/abs/2304.01262 and https://link.springer.com/article/10.1007/JHEP10(2023)053 in line 68

Reply: Thanks for bringing the references to our attention. We have cited the first reference properly. We also have cited the second reference in line 68. Nevertheless, please note that our preprint was posted on arXiv in January 2023 (a version very close to the one submitted SciPost), prior to the publication of these papers.

10) Regarding outlook point 2, I'm confused about why you need the $x_i$ to come from other module categories. You already seem to get access to different phases just by tuning the ${w^z_h}$ variables? And the reason for changing the module category is to access other 1+1D phases with the same graded fusion category symmetry. Does this mean that for a given set of ${w^z_h}$ for a fixed set ${x_i}$ you only get access to a single gapped phase (and gapless regions)? And in order to access other gapped phases you need to change ${x_i}$? For example, for the simple $G=Z_2$ case with $C_0={1}$, you get both the trivial symmetric phase ($w^z_h=1$ for all $h,z$) and the SSB phase ($w^z_0=1,w^z_{h\neq 0}=0$). Can you comment more on which phases you expect to access given ${x_i}$?

Reply: Indeed, by tuning $ {w_{h}^z} $, we are already able to access different gapped/gapless phases. The purpose of the suggestion to make use of module categories is to have a more general way to construct 1D lattice models (note that $\mathcal{C}_G$ is a module category of $\mathcal{C}_G$ itself). As a comparison, let us take the spin-$\frac{1}{2}$ and spin-1 chains with $ SO(3) $ symmetry as examples. Spin $ \frac{1}{2}$'s and spin 1's realize different representations of $ SO(3)$. As is well-known, the spin-$ \frac{1}{2} $ chain and spin-1 chain host very different physics, even though they share the same $ SO(3) $ symmetry. Therefore, generally speaking, having a more general way to construct models is always useful. Nevertheless, neither do we claim that the current construction has a severe limitation in accessing phases of matter with fusion category symmetry, nor do we suggest that the general construction with module categories can give rise to new gapless phases of matter.

11) Regarding outlook point 3, I'm not sure what you mean by maximal category symmetry here? For example, you write "More generally, one may expect a larger category symmetry $\mathcal{Z}(\mathcal{C}_G)$ in the gapless state of the model." However, $\mathcal{Z}(\mathcal{C}_G)$ is braided while the symmetry of a 1+1D system contains only fusion data? It seems that the maximal category symmetry should be another fusion category, not the center of the fusion category which is a braided tensor category?

Reply: By ``maximal category symmetry", we mean the category $ \mathcal{Z}(\mathcal{C}_G) $. It describes the bulk topological order after gauging $G$ in the bulk SET. Generally speaking, any closed string operator corresponding to moving a bulk anyon (a simple object in $\mathcal{Z}(\mathcal{C}_G)$) along the full circle of the boundary of the disk geometry can be viewed as a symmetry of the boundary theory. Accordingly, the full set of symmetries should be described by $\mathcal{Z}(\mathcal{C}_G)$. Nevertheless, it is not easy implement the full symmetries. Taking the boundary of the toric-code topological order for example, the four anyons $1,e,m,\epsilon$ gives rise to a $Z_2^e\times Z_2^m$ symmetry. The current 1D construction only makes explicit use of $Z_2^m$, generated by moving $m$. The $Z_2^e$, related to $Z_2^m$ by the Kramers-Wannier duality, is not implemented in our construction. If the full $Z_2^e\times Z_2^m$ can be made use of, the model shall be pinned at the self-dual point under Kramers-Wannier duality. That means, a larger symmetry is implemented if one can make the full use of $\mathcal{Z}(\mathcal{C}_G)$.

We borrowed the terminology ``maximal category symmetry'' from Ref. 60 (the previous version of the manuscript), which is defined for a particular CFT under consideration. We realize that it might not be appropriate in our context, as we do not have a specific CFT in mind. Accordingly, we have removed this terminology and rephrased the Outlook Point 3 in the revised manuscript. We thank the referee for bringing up the issue.

12)You don't have to do this, but it would be interesting to compare the anomalous vs non-anomalous phase diagram, to make the claim about "larger likelihood to be gapless" more concrete. Here you only have the phase diagram in Fig 3 for the model with the anomalous Z2 symmetry.

Reply: This is a great suggestion. We thank the referee for bringing this up. We have included a new Fig. 3 (phase diagram of $H^0$) as a comparison to Fig. 4 (phase diagram of $H^1$).

---

## Round 2 · Referee Report · Anonymous (Referee 1) · 2024-9-23

Report

I am happy with the authors' response. I think the paper can be published as it is now.

Recommendation

Publish (easily meets expectations and criteria for this Journal; among top 50%)

---

## Round 2 · Author Response

Dear Editor,

We thank the referees for very positive reports. Both of them consider our work of high validity and originality. Referee 1 only asks a minor question about the relation of our work to another work in the literature. While Referee 2 asks many questions, they are all about conceptual or technical clarifications, or about presentation issues of the manuscript. We have replied to all the questions below, and have revised the manuscript according to the suggestions of the referees. We hope the revised manuscript is ready for publication in SciPost.

Yours Sincerely,

Shangqiang Ning, Bin-Bin Mao and Chenjie Wang

---

## Round 2 · List of Changes

1. Added the phase diagram of $H^0$ in Fig. 3.
  2. Revised a few paragraphs of Section 3.2.1 due to the addition of the phase diagram of $H^0$.
  3. Minor revisions of the presentation in several places suggested by Referees

---

## Editorial Decision

published